# The Alk receptor tyrosine kinase regulates Sparkly, a novel activity regulating neuropeptide precursor in the *Drosophila* central nervous system

Sanjay Kumar Sukumar[1], Vimala Antonydhason[1], Linnea Molander[1], Jawdat Sandakly[2], Malak Kleit[2], Ganesh Umapathy[1], Patricia Mendoza-Garcia[1], Tafheem Masudi[1], Andreas Schlosser[3], Dick R Nässel[4], Christian Wegener[5], Margret Shirinian[2], Ruth H Palmer[1]*

[1]Department of Medical Biochemistry and Cell Biology, Institute of Biomedicine, University of Gothenburg, Gothenburg, Sweden; [2]Department of Experimental Pathology, Immunology and Microbiology, Faculty of Medicine, American University of Beirut, Beirut, Lebanon; [3]Julius-Maximilians-Universität Würzburg, Rudolf-Virchow-Center, Center for Integrative and Translational Bioimaging, Würzburg, Germany; [4]Department of Zoology, Stockholm University, Stockholm, Sweden; [5]Julius-Maximilians-Universität Würzburg, Biocenter, Theodor-Boveri-Institute, Neurobiology and Genetics, Würzburg, Germany

*For correspondence:
ruth.palmer@gu.se

Competing interest: The authors declare that no competing interests exist.

**Abstract** Numerous roles for the Alk receptor tyrosine kinase have been described in *Drosophila*, including functions in the central nervous system (CNS), however the molecular details are poorly understood. To gain mechanistic insight, we employed Targeted DamID (TaDa) transcriptional profiling to identify targets of Alk signaling in the larval CNS. TaDa was employed in larval CNS tissues, while genetically manipulating Alk signaling output. The resulting TaDa data were analyzed together with larval CNS scRNA-seq datasets performed under similar conditions, identifying a role for Alk in the transcriptional regulation of neuroendocrine gene expression. Further integration with bulk and scRNA-seq datasets from larval brains in which Alk signaling was manipulated identified a previously uncharacterized *Drosophila* neuropeptide precursor encoded by *CG4577* as an Alk signaling transcriptional target. *CG4577*, which we named *Sparkly (Spar)*, is expressed in a subset of Alk-positive neuroendocrine cells in the developing larval CNS, including circadian clock neurons. In agreement with our TaDa analysis, overexpression of the *Drosophila* Alk ligand Jeb resulted in increased levels of Spar protein in the larval CNS. We show that Spar protein is expressed in circadian (clock) neurons, and flies lacking Spar exhibit defects in sleep and circadian activity control. In summary, we report a novel activity regulating neuropeptide precursor gene that is regulated by Alk signaling in the *Drosophila* CNS.

## eLife assessment

This paper characterises a novel gene (*Spar*), and presenting **valuable** findings in the field of insect biology and behaviour. The experiments are well designed, with attention to detail, showcasing the potential of the *Drosophila melanogaster* model and the use of online resources. The mixed approach presents a **convincing** argument for a genetic interaction between Alk and *Spar*.

## Introduction

Receptor tyrosine kinases (RTK) are involved in a wide range of developmental processes. In humans, the Aanaplastic Lymphoma inase (ALK) RTK is expressed in the central and peripheral nervous system and its role as an oncogene in the childhood cancer neuroblastoma, which arises from the peripheral nervous system, is well described (*Iwahara et al., 1997*; *Matthay et al., 2016*; *Umapathy et al., 2019*; *Vernersson et al., 2006*).

In *Drosophila melanogaster,* Alk is expressed in the visceral mesoderm, central nervous system (CNS), and at neuromuscular junctions (NMJ). The critical role of *Drosophila* Alk and its ligand Jelly belly (Jeb) in the development of the embryonic visceral mesoderm has been extensively studied (*Englund et al., 2003*; *Jin et al., 2013*; *Lee et al., 2003*; *Lorén et al., 2003*; *Mendoza-Garcia et al., 2021*; *Mendoza-García et al., 2017*; *Pfeifer et al., 2022*; *Popichenko et al., 2013*; *Reim et al., 2012*; *Schaub and Frasch, 2013*; *Shirinian et al., 2007*; *Stute et al., 2004*; *Varshney and Palmer, 2006*; *Wolfstetter et al., 2017*). In the CNS, Alk signaling has been implicated in diverse functions, including targeting of photoreceptor axons in the developing optic lobes (*Bazigou et al., 2007*), regulation of NMJ synaptogenesis and architecture (*Rohrbough and Broadie, 2010*; *Rohrbough et al., 2013*), and mushroom body neuronal differentiation (*Pfeifer et al., 2022*). In addition, roles for Alk in neuronal regulation of growth and metabolism, organ sparing and proliferation of neuroblast clones, as well as sleep and long-term memory formation in the CNS have been reported (*Bai and Sehgal, 2015*; *Cheng et al., 2011*; *Gouzi et al., 2011*; *Orthofer et al., 2020*). The molecular mechanisms underlying these Alk-driven phenotypes are currently under investigation, with some molecular components of *Drosophila* Alk signaling in the larval CNS, such as the protein tyrosine phosphatase Corkscrew, identified in recent BioID-based in vivo proximity labeling analyses (*Uçkun et al., 2021*).

In this work, we aimed to capture Alk signaling-dependent transcriptional events in the *Drosophila* larval CNS using Targeted DamID (TaDa) that profiles RNA polymerase II (Pol II) occupancy. TaDa employs a prokaryotic DNA adenine methyltransferase (Dam) to specifically methylate adenines within GATC sequences present in the genome, creating unique GA$^{me}$TC marks. In TaDa, expression of Dam fused to Pol II results in GA$^{me}$TC marks on sequences adjacent to the Pol II binding site and can be combined with the Gal4/UAS system to achieve cell type-specific transcriptional profiling (*Southall et al., 2013*). Tissue-specific TaDa analysis of Alk signaling, while genetically manipulating Alk signaling output, has previously been used to identify Alk transcriptional targets in the embryonic visceral mesoderm, such as the transcriptional regulator *Kahuli* (*Mendoza-Garcia et al., 2021*). Here, we employed this strategy to identify Alk transcriptional targets in *Drosophila* larval brain tissue. These Alk TaDa-identified transcripts were enriched in neuroendocrine cells. Further integration with bulk RNA-seq datasets generated from *Alk* gain-of-function and loss-of-function alleles identified the uncharacterized neuropeptide precursor (*CG4577*), as an Alk target in the *Drosophila* brain, that we have named *Sparkly (Spar)* based on its protein expression pattern. Spar is expressed in a subset of Alk-expressing cells in the central brain and ventral nerve cord, overlapping with the expression pattern of the neuroendocrine-specific transcription factor Dimmed (Dimm) (*Hewes et al., 2003*). Further, using genetic manipulation of Alk we show that Spar levels in the CNS respond to Alk signaling output, validating *Spar* as a transcriptional target of Alk. *Spar* mutant flies showed significant reduction in lifespan, and behavioral phenotypes including defects in activity, sleep, and circadian activity. Notably, *Alk* loss-of-function alleles displayed similar behavioral defects, suggesting that Alk-dependent regulation of Spar in peptidergic neuroendocrine cells modulates activity and sleep/rest behavior. Interestingly, Alk and its ligand Alkal2 play a role in regulation of behavioral and neuroendocrine function in vertebrates (*Ahmed et al., 2022*; *Bilsland et al., 2008*; *Borenäs et al., 2021*; *Lasek et al., 2011a*; *Lasek et al., 2011b*; *Orthofer et al., 2020*; *Weiss et al., 2012*; *Witek et al., 2015*). Taken together, our findings suggest an evolutionarily conserved role of Alk signaling in the regulation of neuroendocrine cell function and identify *Spar* as the first molecular target of Alk to be described in the regulation of activity and circadian control in the fly.

## Results

### TaDa identifies Alk-regulated genes in *Drosophila* larval CNS

To characterize Alk transcriptional targets in the *Drosophila* CNS, we employed TaDa. Briefly, transgenic Dam fused with RNA-Pol II (hereafter referred as Dam-Pol II) (*Southall et al., 2013*; *Figure 1a*

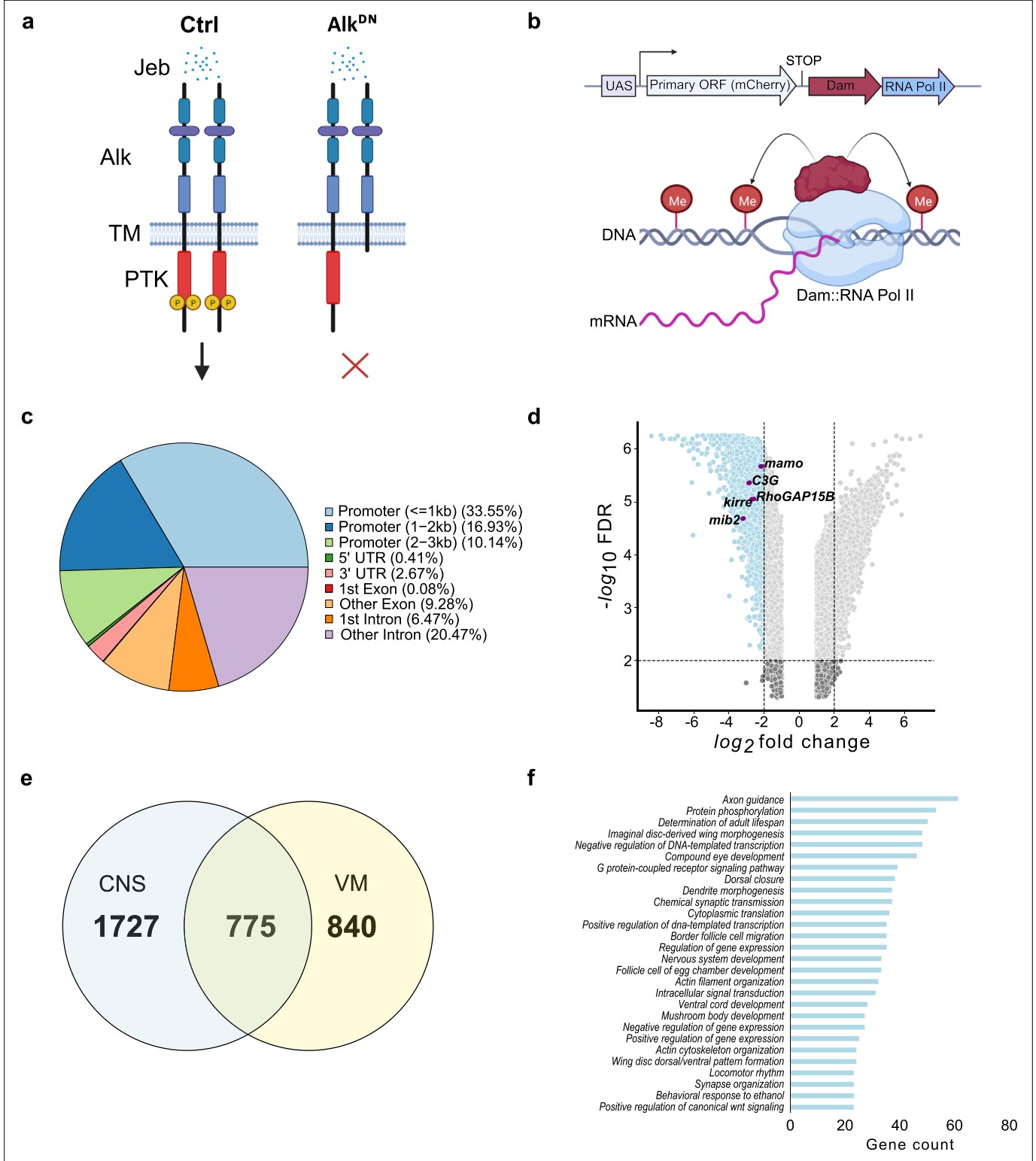

**Figure 1.** Targeted DamID (TaDa)-seq identifies novel Alk-regulated genes in the *Drosophila* larval central nervous system (CNS). (**a**) Schematic overview of experimental conditions comparing wild-type Alk (Ctrl) with Alk dominant-negative (*Alk^DN*) conditions. The *Drosophila* Alk receptor tyrosine kinase (RTK) is comprised of extracellular, transmembrane and intracellular kinase (red) domains. Upon Jelly belly (Jeb, blue dots) ligand stimulation the Alk kinase domain is auto-phosphorylated (yellow circles) and downstream signaling is initiated. In *Alk^DN* experimental conditions, Alk signaling is inhibited due to the overexpression of the Alk extracellular domain. (**b**) The TaDa system (expressing *Dam::RNA Pol II*) leads to the methylation of GATC sites in the genome, allowing transcriptional profiling based on RNA Pol II occupancy. (**c**) Pie chart indicating the distribution of TaDa peaks on various genomic

*Figure 1 continued on next page*

*Figure 1 continued*

features such as promoters, 5' UTRs, 3' UTRs, exons, and introns. (**d**) Volcano plot of TaDa-positive loci enriched in *Alk^DN* experimental conditions compared to control loci exhibiting log2FC≥2, p≤0.05 are shown in blue. Alk-associated genes such as *mamo, C3G, Kirre, RhoGAP15B,* and *mib2* are highlighted in purple. (**e**) Venn diagram indicating Alk-dependent TaDa downregulated genes from the current study compared with the previously identified Alk-dependent TaDa loci in the embryonic VM (*Mendoza-Garcia et al., 2021*). (**f**) Enrichment of Gene Ontology (GO) terms associated with significantly downregulated genes in *Alk^DN* experimental conditions. Illustrations in *Figure 1a* and a portion of *Figure 1b* were created with BioRender. com, and published using a CC BY-NC-ND license with permission.

The online version of this article includes the following figure supplement(s) for figure 1:

**Figure supplement 1.** Targeted DamID (TaDa) third instar larval central nervous system (CNS) sample validation and additional data analysis.

*and b*) was driven using the pan neuronal *C155-Gal4* driver. To inhibit Alk signaling we employed a dominant-negative Alk transgene, which encodes the Alk extracellular and transmembrane domain (hereafter referred as *UAS-Alk^DN*) (*Bazigou et al., 2007*; *Figure 1a*). Flies expressing Dam-Pol II alone in a wild-type background were used as control. Expression of Dam-Pol II was confirmed by expression of mCherry, which is encoded by the primary ORF of the TaDa construct (*Southall et al., 2013*; *Figure 1b*, *Figure 1—figure supplement 1a and b'*). CNS from third instar wandering larvae were dissected and genomic DNA was extracted, fragmented at GA^meTC marked sites using methylation-specific DpnI restriction endonuclease. The resulting GATC fragments were subsequently amplified for library preparation and NGS sequencing (*Figure 1—figure supplement 1c*). Bioinformatic data analysis was performed based on a previously described pipeline (*Marshall and Brand, 2017*; *Mendoza-Garcia et al., 2021*). Initial quality control analysis indicated comparable numbers of quality reads between samples and replicates, identifying >20 million raw reads per sample that aligned to the *Drosophila* genome (*Figure 1—figure supplement 1d–d'*). No significant inter-replicate variability was observed (*Figure 1—figure supplement 1e*). Meta-analysis of reads associated with GATC borders showed a tendency to accumulate close to transcription start sites (TSS) indicating the ability of TaDa to detect transcriptionally active regions (*Figure 1—figure supplement 1f*). A closer look at the Pol II occupancy profile of *Alk* shows a clear increase in Pol II occupancy from exon 1 to exon 7 (encoding the extracellular and transmembrane domain) in *Alk^DN* samples reflecting the expression of the dominant-negative *Alk* transgene (*Figure 1—figure supplement 1g*).

To detect differential Pol II occupancy between Dam-Pol II control (*C155-Gal4>UAS-LT3-Dam::Pol II*) and *UAS-Alk^DN* (*C155-Gal4>UAS-LT3-Dam::Pol II; UAS-Alk^DN*) samples, neighboring GATC-associated reads, maximum 350 bp apart (median GATC fragment distance in the *Drosophila* genome) were clustered in peaks (*Tosti et al., 2018*). More than 10 million reads in both control and *Alk^DN* samples were identified as GATC-associated reads (*Figure 1—figure supplement 1d'*), and those loci displaying differential Pol II occupancy were defined by logFC and FDR (as detailed in Materials and methods). Greater than 50% of aligned reads were in promoter regions, with 33.55% within a 1 kb range (*Figure 1c*, *Supplementary file 1*).

To further analyze transcriptional targets of Alk signaling, we focused on loci exhibiting decreased Pol II occupancy when compared with controls, identifying 2502 loci with logFC≥2, FDR≤0.05 (*Figure 1d*, *Supplementary file 1*). Genes previously known to be associated with Alk signaling, such as *kirre, RhoGAP15B, C3G, mib2,* and *mamo*, were identified among downregulated loci (*Figure 1d*). We compared CNS TaDa Alk targets with our previously published embryonic visceral mesoderm TaDa datasets that were derived under similar experimental conditions (*Mendoza-Garcia et al., 2021*) and found 775 common genes (*Figure 1e*, *Supplementary file 1*). Gene Ontology (GO) analysis identified GO terms in agreement with previously reported Alk functions in the CNS (*Bai and Sehgal, 2015*; *Bazigou et al., 2007*; *Cheng et al., 2011*; *Gouzi et al., 2011*; *Orthofer et al., 2020*; *Pfeifer et al., 2022*; *Rohrbough and Broadie, 2010*; *Rohrbough et al., 2013*; *Woodling et al., 2020*) such as axon guidance, determination of adult lifespan, nervous system development, regulation of gene expression, mushroom body development, behavioral response to ethanol and locomotor rhythm (*Figure 1f*). Many of the differentially regulated identified loci have not previously been associated with Alk signaling and represent candidates for future characterization.

## TaDa targets are enriched for neuroendocrine transcripts

To further characterize Alk-regulated TaDa loci, we set out to examine their expression in scRNA-seq data from wild-type third instar larval CNS (*Pfeifer et al., 2022*). Enrichment of TaDa loci were identified by using AUCell, an area-under-the-curve-based enrichment score method, employing the top 500 TaDa hits (*Aibar et al., 2017*; *Figure 2a and b*, *Supplementary file 1*). This analysis identified 786 cells (out of 3598), mainly located in a distinct cluster of mature neurons that robustly express both *Alk* and *jeb* (*Figure 2b*, red circle; *Figure 2c*). This cluster was defined as neuroendocrine cells based on canonical markers, such as the neuropeptides *Lk* (*Leucokinin*), *Nplp1* (*Neuropeptide-like precursor 1*), *Dh44* (*Diuretic hormone 44*), *Dh31* (*Diuretic hormone 31*), *sNPF* (*short neuropeptide F*), *AstA* (*Allatostatin A*), and the enzyme *Pal2* (*Peptidyl-α-hydroxyglycine-α-amidating lyase 2*) as well as *Eip74EF* (*Ecdysone-induced protein 74EF*), and *Rdl* (resistance to dieldrin) (*Guo et al., 2019*; *Hückesfeld et al., 2021*; *Takeda and Suzuki, 2022*; *Torii, 2009*; *Figure 2d–f*). Overall, the TaDa-scRNA-seq data integration analysis suggests a role of Alk signaling in regulation of gene expression in neuroendocrine cells.

To further explore the observed enrichment of Alk-regulated TaDa loci in neuroendocrine cells, we used a Dimm transcription factor reporter (*Dimm-Gal4>UAS-GFPcaax*), as a neuroendocrine marker (*Park et al., 2008*), to confirm Alk protein expression in a subset of neuroendocrine cells in the larval central brain, ventral nerve cord, and neuroendocrine corpora cardiaca cells (*Figure 2g*, *Figure 2—figure supplement 1*). This could not be confirmed at the RNA level, due to low expression of *dimm* in both our and publicly available single-cell RNA-seq datasets (*Brunet Avalos et al., 2019*; *Michki et al., 2021*; *Pfeifer et al., 2022*).

## Multi-omics integration identifies *CG4577* as an Alk transcriptional target

Loci potentially subject to Alk-dependent transcriptional regulation were further refined by integration of the Alk-regulated TaDa dataset with previously collected RNA-seq datasets (*Figure 3a*). Specifically, *w^1118^* (control), *Alk^Y1355S^* (Alk gain-of-function), and *Alk^ΔRA^* (Alk loss-of-function) RNA-seq datasets (*Pfeifer et al., 2022*) were compared to identify genes that exhibited both significantly increased expression in Alk gain-of-function conditions (*w^1118^* vs *Alk^Y1355S^*) and significantly decreased expression in Alk loss-of-function conditions (*w^1118^* vs *Alk^ΔRA^* and control vs *C155-Gal4*-driven expression *of UAS-Alk^DN^*). Finally, we positively selected for candidates expressed in *Alk*-positive cells in our scRNA-seq dataset. Notably, the only candidate which met these stringent criteria was *CG4577*, which encodes an uncharacterized putative neuropeptide precursor (*Figure 3b*). *CG4577* exhibited decreased Pol II occupancy in *Alk^DN^* samples (*Figure 3c*), and *CG4577* transcripts were upregulated in *Alk^Y1355S^* gain-of-function conditions and downregulated in *Alk^ΔRA^* loss-of-function conditions (*Figure 3d*, *Supplementary file 1*). In agreement with a potential role as a neuropeptide precursor, expression of *CG4577* was almost exclusively restricted to neuroendocrine cell clusters in our scRNA-seq dataset (*Figure 3e*). Examination of additional publicly available first instar larval and adult CNS scRNA-seq datasets (*Brunet Avalos et al., 2019*; *Davie et al., 2018*) confirmed the expression of *CG4577* in Alk-expressing cells (*Figure 3—figure supplement 1a and b*). *CG4577-RA* encodes a 445 amino acid prepropeptide with a 27 amino acid N-terminal signal peptide sequence as predicted by SignalP-5.0 (*Figure 3f*; *Almagro Armenteros et al., 2019*). Analysis of CG4577-PA at the amino acid level identified a high percentage of glutamine residues (43 of 445; 9%), including six tandem glutamine repeats (amino acids 48–56, 59–62, 64–71, 116–118, 120–122, and 148–150) of unknown function as well as a lack of cysteine residues. The preproprotein has an acidic pI of 5.1 and carries a net negative charge of 6. Several poly- and dibasic proprotein convertase (PC) (proprotein convertase) cleavage sites were also predicted (KR, KK, RR, RK) (*Pauls et al., 2014*; *Southey et al., 2006*; *Veenstra, 2000*; *Figure 3f*). Since the propeptide does not contain cysteine residues, it is unable to form intracellular or dimeric disulfide bridges. A second transcript, *CG4577-RB*, encodes a 446 amino acid protein with only two amino acid changes (*Figure 3—figure supplement 1c*). Phylogenetic analysis of *CG4577* relative to known *Drosophila* neuropeptide precursors failed to identify strong homology in keeping with the known low sequence conservation of neuropeptide prepropeptides outside their bioactive peptide stretches. However, we were also unable to find sequence homologies with other known invertebrate or vertebrate peptides. Next, we searched for *CG4577* orthologs across Metazoa. We obtained orthologs across the Drosophilids, Brachyceran flies, and Dipterans. No orthologs were found at higher

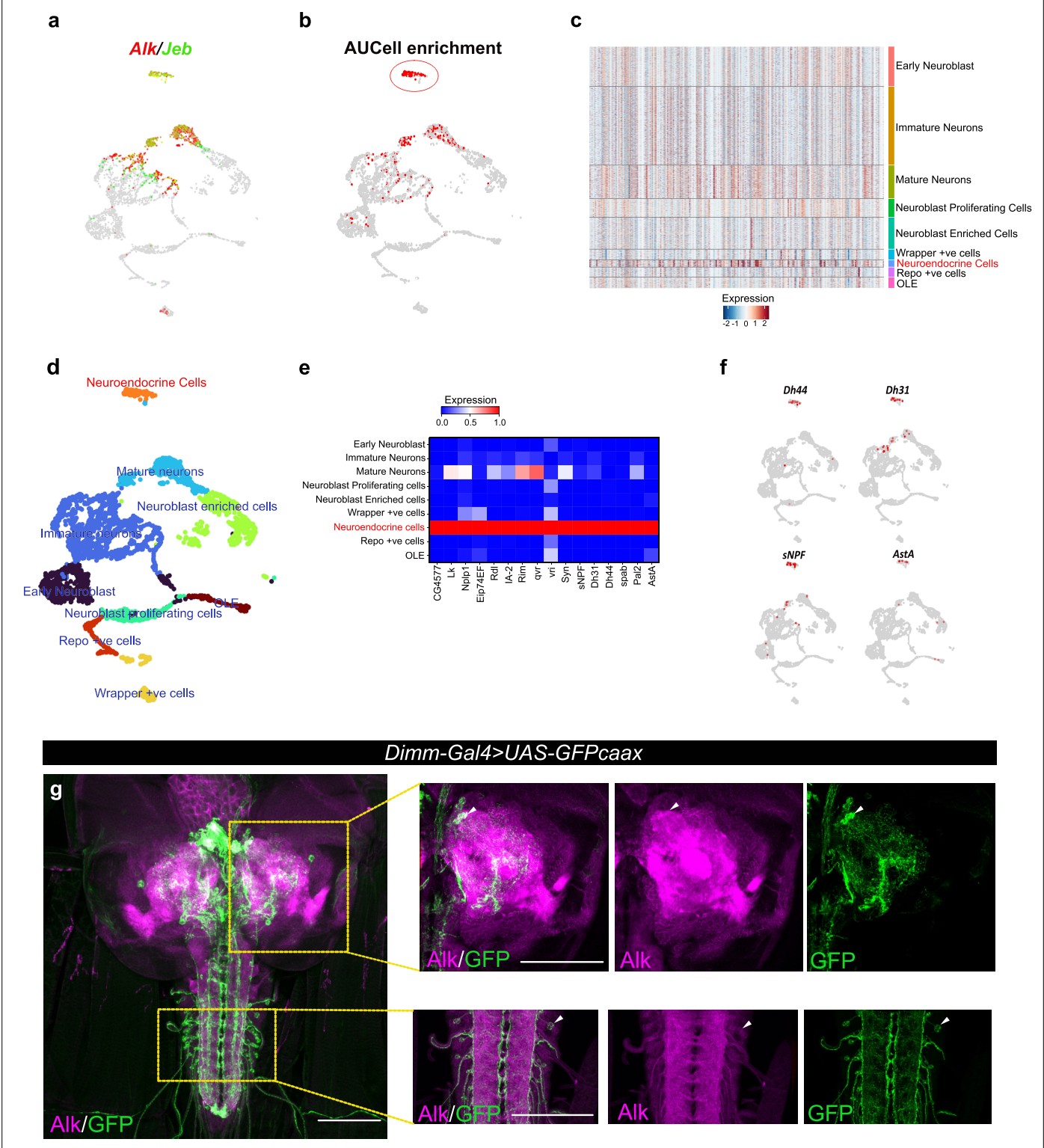

**Figure 2.** Integration of Targeted DamID (TaDa) data with scRNA-seq identifies an enrichment of Alk-regulated genes in neuroendocrine cells. (**a**) Uniform Manifold Approximation and Projection (UMAP) feature plot indicating *Alk* (in red) and *Jeb* (in green) mRNA expression in a control (*w^1118*) third instar larval central nervous system (CNS) scRNA-seq dataset (*Pfeifer et al., 2022*). (**b**) UMAP visualizing AUCell enrichment analysis of the top 500 TaDa downregulated genes in the third instar larval CNS scRNA-seq dataset. Cells exhibiting an enrichment (threshold>0.196) are depicted in red. One highly enriched cell cluster is highlighted (red circle). (**c**) Heatmap representing expression of the top 500 genes downregulated in TaDa *Alk^DN* samples across larval CNS scRNA-seq clusters identifies enrichment in neuroendocrine cells. (**d**) UMAP indicating third instar larval CNS annotated clusters (*Pfeifer*

*Figure 2 continued on next page*

*Figure 2 continued*

*et al., 2022*), including the annotated neuroendocrine cell cluster (in orange). (**e**) Matrix plot displaying expression of canonical neuroendocrine cell markers. (**f**) UMAPs visualizing mRNA expression of *Dh44*, *Dh31*, *sNPF*, and *AstA* neuropeptides across the scRNA population. (**g**) Alk staining in *Dimm-Gal4>UAS-GFPcaax* third instar larval CNS confirms Alk expression in Dimm-positive cells. Alk (in magenta) and GFP (in green), close-ups indicated by boxed regions and arrows indicating overlapping cells in the central brain and ventral nerve cord. Scale bars: 100 μm.

The online version of this article includes the following figure supplement(s) for figure 2:

**Figure supplement 1.** Alk is expressed in Dimm-Gal4>UAS-GFPcaax positive cells of the third instar larval CNS.

**Figure supplement 2.** Feature plots visualizing expression of Targeted DamID (TaDa)-identified genes expressed in neuroendocrine cells in scRNA-seq from third instar larval central nervous system (CNS) (*Pfeifer et al., 2022*).

taxonomic levels, suggesting that *CG4577* either originated in Dipterans or has a high sequence variability at higher taxonomic levels. To identify conserved peptide stretches indicating putative bioactive peptide sequences, we aligned the predicted amino acid sequences of the Dipteran *CG4577* orthologs. This revealed several conserved peptide stretches (*Figure 3—figure supplement 2*) framed by canonical PC cleavage sites that might represent bioactive peptide sequences. BLAST searches against these conserved sequences did not yield hits outside of the Diptera.

## CG4577/Spar is expressed in neuroendocrine cells

To further characterize *CG4577* we generated polyclonal antibodies that are predicted to recognize both CG4577-PA and CG4577-PB and investigated protein expression. CG4577 protein was expressed in a 'sparkly' pattern in neurons of the third instar central brain as well as in distinct cell bodies and neuronal processes in the ventral nerve cord, prompting us to name CG4577 as Sparkly (Spar) (*Figure 4a and b*). Co-labeling of Spar and Alk confirmed the expression of Spar in a subset of Alk-expressing cells, in agreement with our transcriptomics analyses (*Figure 4a*, *Figure 4—figure supplement 1*). In addition, we also observed expression of Spar in neuronal processes which emerge from the ventral nerve cord and appear to innervate larval body wall muscle number 8, that may be either Leukokinin (Lk) or cystine-knot glycoprotein hormone GPB5 expressing neurons (*Figure 4b*; *Cantera and Nässel, 1992*; *Sellami et al., 2011*). Spar antibody specificity was confirmed in both *C155-Gal4>UAS-Spar-RNAi* larvae, where RNAi-mediated knockdown of *Spar* resulted in loss of detectable signal (*Figure 4c–c'*), and in *C155-Gal4>UAS* Spar larvae, exhibiting ectopic *Spar* expression in the larval CNS and photoreceptors of the eye disc (*Figure 4d–d'*). To further address Spar expression in the neuroendocrine system, we co-labeled with antibodies against Dimm to identify peptidergic neuronal somata (*Allan et al., 2005*) in a *Dimm-Gal4>UAS-GFPcaax* background. This further confirmed the expression of Spar in Dimm-positive peptidergic neuroendocrine cells in the larval CNS (*Figure 4e–e''*, *Figure 4—videos 1 and 2*). Moreover, co-staining of Spar and Dimm in the adult CNS showed similar results (*Figure 4—figure supplement 2*).

## Spar expression is modulated in response to Alk signaling activity

Our initial integrated analysis predicted *Spar* as a locus responsive to Alk signaling. To test this hypothesis, we examined Spar protein expression in $w^{1118}$, $Alk^{Y1355S}$, and $Alk^{\Delta RA}$ genetic backgrounds, in which Alk signaling output is either upregulated ($Alk^{Y1355S}$) or downregulated ($Alk^{\Delta RA}$) (*Pfeifer et al., 2022*). We observed a significant increase in Spar protein in $Alk^{Y1355S}$ CNS, while levels of Spar in $Alk^{\Delta RA}$ CNS were not significantly altered (*Figure 4f–h*, quantified in i). In agreement, overexpression of Jeb (*C155-Gal4>jeb*) significantly increased Spar levels when compared with controls (*C155-Gal4>UAS-GFPcaax*) (*Figure 4j–l*, quantified in m). Again, overexpression of dominant-negative Alk (*C155-Gal4>UAS-Alk^{DN}*) did not result in significantly decreased Spar levels (*Figure 4l*, quantified in m). Thus activation of Alk signaling increases Spar protein levels. However, while our bulk RNA-seq and TaDa datasets show a reduction in *Spar* transcript levels in Alk loss-of-function conditions, this reduction is not reflected at the protein level. This observation may reflect additional uncharacterized pathways that regulate *Spar* mRNA levels as well as translation and protein stability, since and notably *Spar* transcript levels are decreased but not absent in $Alk^{\Delta RA}$ (*Figure 3d*). Taken together, these observations confirm that *Spar* expression is responsive to Alk signaling in CNS, although Alk is not critically required to maintain Spar protein levels.

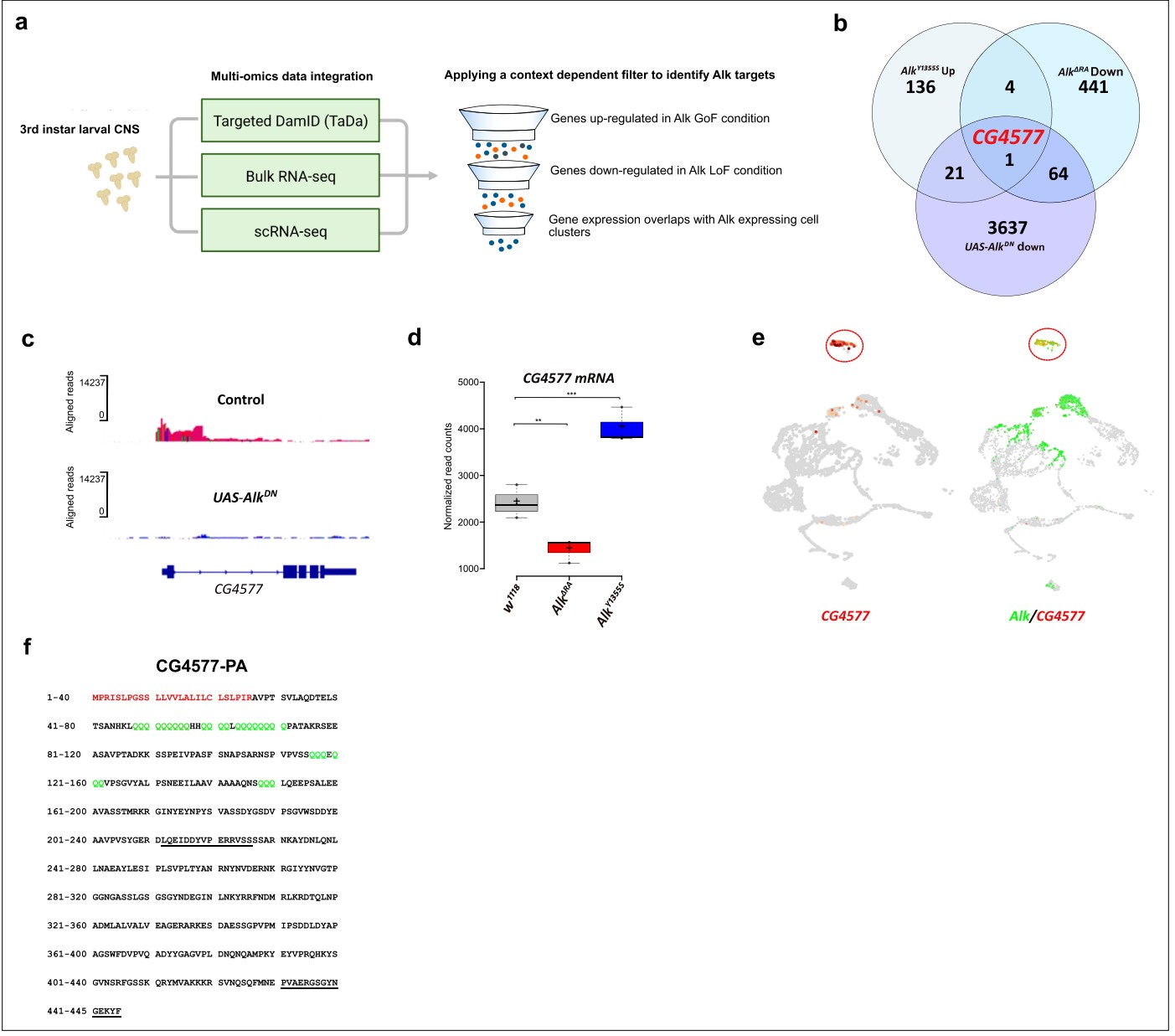

**Figure 3.** Targeted DamID (TaDa) and RNA-seq identifies CG4577 as a novel Alk-regulated neuropeptide. (**a**) Flowchart representation of the multi-omics approach employed in the study and the context-dependent filter used to integrate TaDa, bulk RNA-seq, and scRNA-seq datasets. (**b**) Venn diagram comparing bulk RNA-seq (log2FC>1.5, p≤0.05) and TaDa datasets (log2FC≥2, p≤0.05). A single candidate (*CG4577/Spar*) was identified as responsive to Alk signaling. (**c**) TaDa Pol II occupancy of *CG4577/Spar* shows decreased occupancy in *Alk^DN* experimental conditions compared to control. (**d**) Expression of *CG4577/Spar* in *w^1118* (control), *Alk^ΔRA* (*Alk* loss-of-function allele), and *Alk^Y1355S* (*Alk* gain-of-function allele) larval central nervous system (CNS). Boxplot with normalized counts, **p<0.01, ***p<0.001. (**e**) Uniform Manifold Approximation and Projections (UMAPs) showing mRNA expression of *CG4577/Spar* and *Alk* in third instar larval CNS scRNA-seq data. Neuroendocrine cluster is highlighted (red circle). (**f**) CG4577/Spar-PA amino acid sequence indicating the signal peptide (amino acids 1–26, in red), glutamine repeats (in green), and the anti-CG4577/Spar antibody epitopes (amino acids 211–225 and 430–445, underlined). Center lines in boxplots indicate medians; box limits indicate the 25th and 75th percentiles; crosses represent sample means; whiskers extend to the maximum or minimum.

The online version of this article includes the following figure supplement(s) for figure 3:

**Figure supplement 1.** Co-expression of *Alk* and *Spar* in publicly available *Drosophila* central nervous system (CNS) scRNA-seq datasets.

**Figure supplement 2.** Alignment of *CG4577* orthologs in flies (Brachycera).

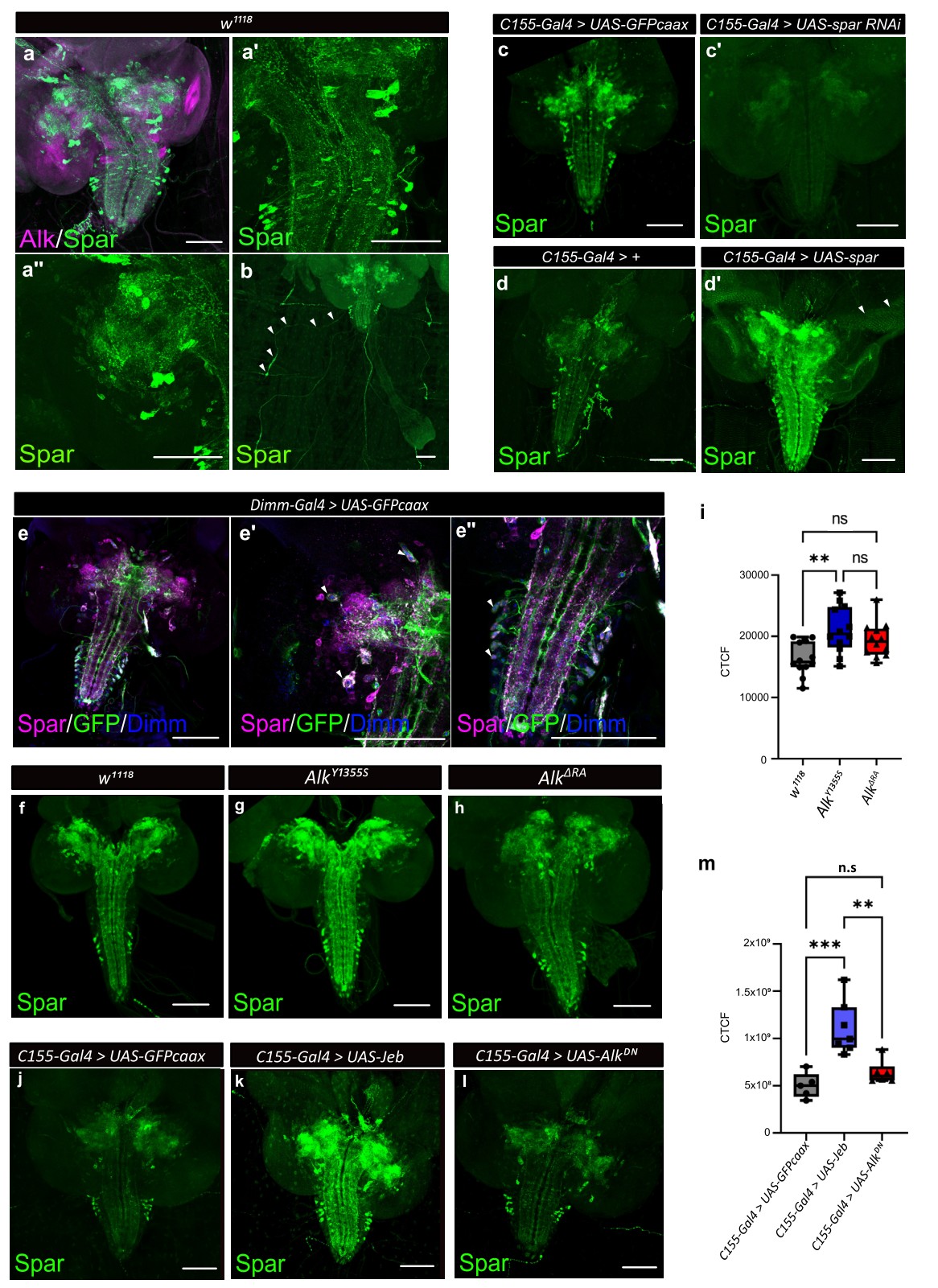

**Figure 4.** Spar expression in the *Drosophila* larval brain. (**a**) Immunostaining of *w[1118]* third instar larval brains with Spar (green) and Alk (magenta) revealing overlapping expression in central brain and ventral nerve cord. (**a'–a''**) Close-up of Spar expression (green) in ventral nerve cord (**a'**) and central brain (**a''**). (**b**) Immunostaining of *w[1118]* third instar larval central nervous system (CNS) together with the body wall muscles, showing Spar (green) expression in neuronal processes (white arrowheads) which emerge from the ventral nerve cord and innervate larval body wall muscle number 8.

*Figure 4 continued on next page*

*Figure 4 continued*

(**c–c′**) Decreased expression of Spar in third instar larval brains expressing *Spar* RNAi (*C155-Gal4>Spar* RNAi) compared to control (*C155-Gal4>UAS-GFPcaax*) confirms Spar antibody specificity (Spar in green). (**d–d′**) Spar overexpression (*C155-Gal4>UAS* Spar) showing increased Spar expression (in green) compared to control (*C155-Gal4>+*) larval CNS. (**e–e″**) Immunostaining of *Dimm-Gal4>UAS-GFPcaax* third instar larval brains with Spar (in magenta), GFP, and Dimm (in blue) confirms Spar expression in Dimm-positive neuroendocrine cells (white arrowheads). Close-up of ventral nerve cord (**e′**) and central brain (**e″**). (**f–i**) Spar protein expression in *w1118*, *AlkY1355S*, and *AlkΔRA* third instar larval brains. Quantification of Spar levels (corrected total cell fluorescence [CTCF]) in **i**. (**j–m**) Overexpression of Jeb in the third instar CNS (*C155-Gal4>UAS* Jeb) leads to increased Spar protein expression compared to controls (*C155-Gal4>UAS-GFPcaax*). Quantification of Spar levels [CTCF] in **m** (**$p<0.01$; ***$p<0.001$). Scale bars: 100 μm. Center lines in boxplots indicate medians; box limits indicate the 25th and 75th percentiles; whiskers extend to the maximum or minimum.

The online version of this article includes the following video and figure supplement(s) for figure 4:

**Figure supplement 1.** Expression of Alk and Spar in the larval CNS and prothoracic gland.

**Figure supplement 2.** Expression of Spar in Dimm-positive neurons of the adult CNS.

**Figure 4—video 1.** Z-stack projection video of *Figure 4e′*.
https://elifesciences.org/articles/88985/figures#fig4video1

**Figure 4—video 2.** Z-stack projection video of *Figure 4e″*.
https://elifesciences.org/articles/88985/figures#fig4video2

## *Spar* encodes a canonically processed neurosecretory protein

To provide biochemical evidence for the expression of Spar, we re-analyzed data from a previous LC-MS peptidomic analysis of brain extracts from 5-day-old male control flies and flies deficient for carboxypeptidase D (dCPD, SILVER) (*Pauls et al., 2019*), an enzyme that removes the basic C-terminal amino acid of peptides originating from PC cleavage of the proprotein. This analysis identified several peptides derived from the Spar propeptide by mass matching in non-digested extracts from genetic control brains (*Figure 5*). These included peptides that are framed by dibasic prohormone cleavage sequences in the propeptide, one of which (SEEASAVPTAD) was also obtained by de novo sequencing (*Figure 5*). This result demonstrates that the Spar precursor is expressed and is processed into multiple peptides by PCs and possibly also other proteases. Analysis of the brain of *svr* mutant flies yielded similar results, but further revealed peptides C-terminally extended by the dibasic cleavage sequence (SEEEASAVPTADKK, FNDMRLKR) (*Figure 5*), thereby confirming canonical PC processing of the Spar propeptide. Of note, the phylogenetically most conserved peptide sequence of the Spar precursor (DTQLNPADMLALVALVEAGERA, *Figure 3—figure supplement 2*) framed by dibasic cleavage sites was among the identified peptides yet occurred only in control but not *svr* mutant brains (*Figure 5*).

Additionally, we performed co-labeling with known *Drosophila* neuropeptides, pigment-dispersing factor (PDF), Dh44, insulin-like peptide 2 (Ilp2), AstA, and Lk, observing Spar expression in subsets of all these populations (*Figure 6*). These included the PDF-positive ventral lateral neuron (LNv) clock neurons (*Figure 6a–b″*), Dh44-positive neurons (*Figure 6c–d″*), a subset of Ilp2 neurons in the central brain (*Figure 6e–f″*) and several AstA-positive neurons in the central brain and ventral nerve cord (*Figure 6g–h″*). We also noted co-expression in some Lk-positive neurons in the central brain and ventral nerve cord, that include the neuronal processes converging on body wall muscle 8 (*Figure 6i–l″*; *Cantera and Nässel, 1992*). Similar Spar co-expression with PDF, Dh44, Ilp2, and AstA was observed in adult CNS (*Figure 6—figure supplement 1*).

## CRISPR/Cas9-generated *Spar* mutants are viable

Since previous reports have shown that Jeb overexpression in the larval CNS results in a small pupal size (*Gouzi et al., 2011*), we measured pupal size on ectopic expression of Spar (*C155-Gal4>Spar*) and *Spar* RNAi (*C155-Gal4>Spar* RNAi), noting no significant difference compared to controls (*C155-Gal4>+* and *C155-Gal4>jeb*) (*Figure 7—figure supplement 1*). These results suggest that Spar may be involved in an additional Alk-dependent function in the CNS. Further, experiments over-expressing Spar did not reveal any obvious phenotypes. To further investigate the function of Spar, we generated a *Spar* loss-of-function allele by CRISPR/Cas9-mediated non-homologous end-joining, resulting in the deletion of a 716 bp region including the *Spar* TSS and exon 1 (hereafter referred as *SparΔExon1*) (*Figure 7a*). Immunoblotting analysis indicated a 35 kDa protein present in the wild-type (*w1118*) controls that was absent in *SparΔExon1* mutant CNS lysates (*Figure 7b*; *Figure 7—source data 1*). The *SparΔExon1* mutant allele was further characterized using immunohistochemistry (*Figure 7c–d′*).

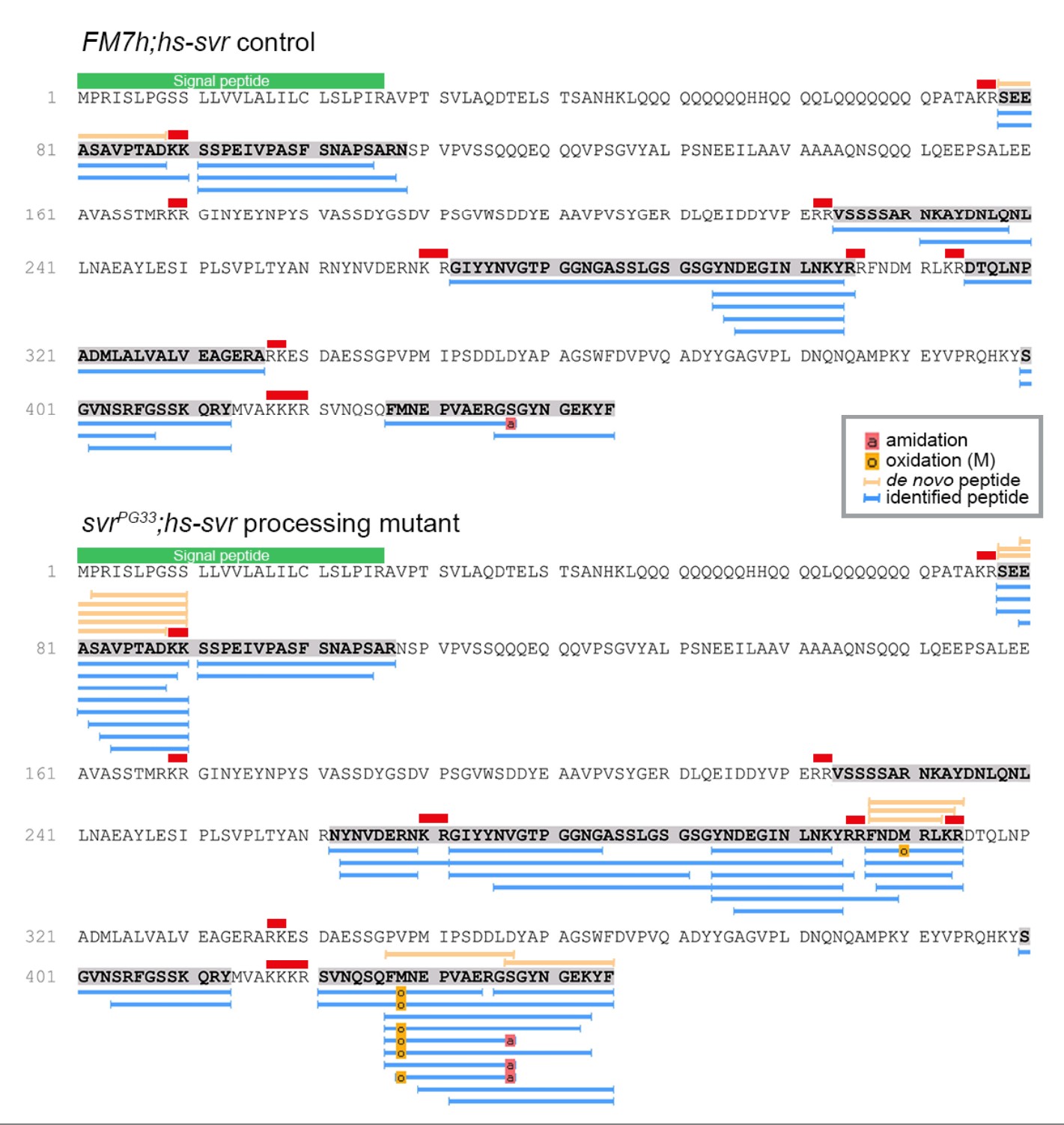

**Figure 5.** Identification of Spar peptides in *Drosophila* central nervous system (CNS) tissues. Peptides derived from the Spar prepropeptide identified by mass spectrometry in wild-type-like control flies (*FM7h;hs-svr*, upper panel) and *svr* mutant (*svrPG33;hs-svr*, lower panel) flies. The predicted amino acid sequence of the CG4577-PA Spar isoform is depicted for each genetic experimental background. Peptides identified by database searching (UniProt *D. melanogaster*, 1% FDR) are marked by blue bars below the sequence. In addition, peptides correctly identified by de novo sequencing are marked by orange bars above the sequence. Red bars indicate basic PC cleavage sites, green bar indicates the signal peptide.

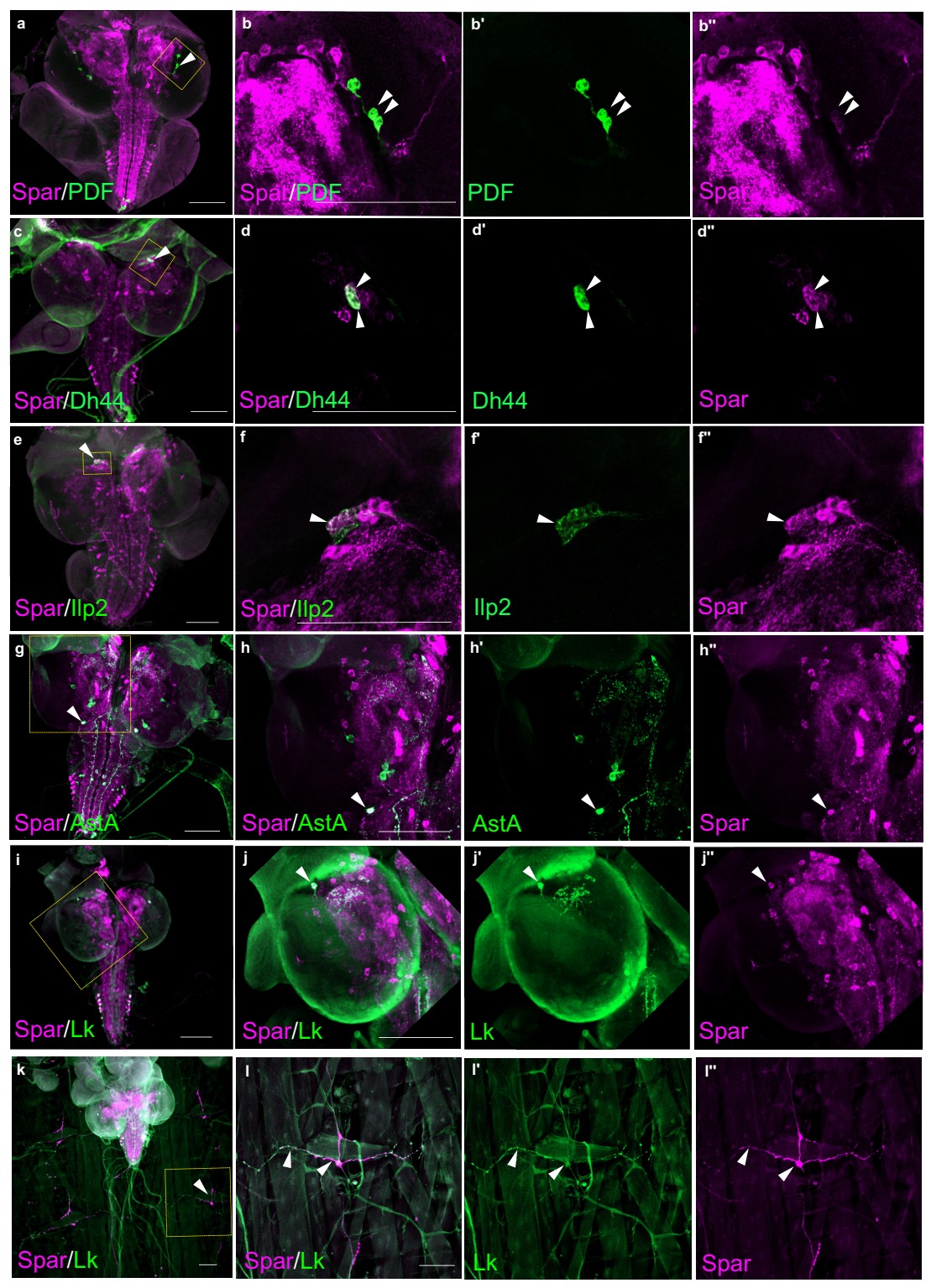

**Figure 6.** *Spar* expression in larval neuropeptide expressing neuronal populations. (**a**) Immunostaining of *w1118* third instar larval central nervous system (CNS) with Spar (in magenta) and PDF (in green). Close-ups (**b–b"**) showing PDF- and Spar-positive neurons in central brain indicated by white arrowheads. (**c**) Immunostaining of *w1118* third instar larval CNS with Spar (in magenta) and Dh44 (in green). Close-ups (**d–d"**) showing Dh44- and Spar-positive neurons in central brain indicated by white arrowheads. (**e**) Immunostaining of *w1118* third instar larval CNS with Spar (in magenta) and Ilp2 (in green). Close-ups (**f–f"**) showing Ilp2- and Spar-positive neurons in central brain indicated by white arrowheads. (**g**) Immunostaining of *w1118* third instar larval CNS with Spar (in magenta) and AstA (in green). Close-ups (**h–h"**) showing AstA- and Spar-positive neurons in central brain indicated by white

*Figure 6 continued on next page*

Figure 6 continued

arrowheads. (**i**) Immunostaining of *w¹¹¹⁸* third instar larval CNS with Spar (in magenta) and Lk (in green). Close-ups (**j–j″**) showing Lk (LHLK neurons)- and Spar-positive neurons in central brain indicated by white arrowheads. (**k**) Immunostaining of *w¹¹¹⁸* third instar larval CNS together with the body wall muscles, showing Spar (in magenta) expressing Lk (in green) (ABLK neurons) in neuronal processes, which emerge from the ventral nerve cord and innervate the larval body wall muscle. Close-ups (**l–l″**) showing co-expression of Lk and Spar in neurons which attach to the body wall number 8 indicated by white arrow heads. Scale bars: 100 µm.

The online version of this article includes the following figure supplement(s) for figure 6:

**Figure supplement 1.** *Spar* expression in adult neuropeptide expressing neuronal populations.

*Spar^ΔExon1^* shows a complete abrogation of larval and adult Spar expression, consistent with the reduction observed when *Spar* RNAi was employed (*Figure 7c–d′*). *Spar^ΔExon1^* flies were viable, and no gross morphological phenotypes were observed, similar to loss of function mutants in several previously characterized neuropeptides such as PDF, drosulfakinin, and neuropeptide F (*Liu et al., 2019*; *Renn et al., 1999*; *Wu et al., 2020*).

## Spar is expressed in a subset of clock neurons in the larval and adult CNS

A previous report noted expression of *Spar* in the LNv, dorsal lateral neuron (LNd), and dorsal neuron 1 (DN1) populations of adult *Drosophila* circadian clock neurons (*Abruzzi et al., 2017*; *Figure 7e*, *Supplementary file 1*). A meta-analysis of the publicly available single-cell transcriptomics of circadian clock neurons indicated that almost all adult clusters of clock neurons express *Spar* (*Ma et al., 2021*; *Figure 7f*). Additionally, we noted that the expression of *Spar* peaks around zeitgeber time 10 (ZT10) (coinciding with the evening peak of locomotor activity) (*Figure 7g and h*), although the differences in expression level around the clock with LD or DD cycle were not dramatic (*Figure 7—figure supplement 2a–c*). To confirm the expression of Spar in circadian neurons at the protein level, we co-stained Spar with a clock neuron reporter (*Clk856-Gal4>UAS* GFP). A subset of Spar-positive larval CNS neurons appeared to be *Clk856-Gal4>UAS* GFP positive (*Figure 7i–j″*). Similarly, a subset of Spar-positive neurons in adults were GFP-positive (*Figure 7k–l″*), confirming the expression of Spar protein in LNv clock neurons. Taken together, these findings suggest a potential function of the Alk-regulated TaDa-identified target Spar in the maintenance of circadian activity in *Drosophila*.

## *Spar^ΔExon1^* mutants exhibit reduced adult lifespan, activity, and circadian disturbances

Given the expression of Spar in circadian neurons of the larval CNS, and the previous observations of a role of Alk mutations in sleep dysregulation in flies (*Bai and Sehgal, 2015*), we hypothesized that *Spar^ΔExon1^* mutants may exhibit activity/circadian rhythm-related phenotypes. To test this, we first investigated the effects of loss of *Spar* (employing *Spar^ΔExon1^*) and loss of *Alk* (employing a CNS-specific loss-of-function allele of *Alk*, *Alk^ΔRA^*, *Pfeifer et al., 2022*) on adult lifespan and sleep/activity behavior using the DAM (*Drosophila* activity monitor) system (Trikinetics Inc). Both *Alk^ΔRA^* and *Spar^ΔExon1^* mutant flies displayed a significantly reduced lifespan when compared to *w¹¹¹⁸* controls, with the *Spar^ΔExon1^* group exhibiting a significant reduction in survival at 25 days (*Figure 8a*). Activity analysis in *Alk^ΔRA^* and *Spar^ΔExon1^* flies under 12 hr light:12 hr dark (LD) conditions indicated that both *Alk^ΔRA^* and *Spar^ΔExon1^* flies exhibited two major activity peaks, the first centered around ZT0, the beginning of the light phase, the so-called morning peak, and the second around ZT12, the beginning of the dark phase that is called the evening peak (*Figure 8b*, black arrows). Overall activity and sleep profiles per 24 hr showed increased activity in *Spar^ΔExon1^* flies (*Figure 8b–d*, *Figure 8—figure supplement 1*), that was more prominent during the light phase, with an increase in the anticipatory activity preceding both the night-day and the day-night transition in comparison to *Alk^ΔRA^* and *w¹¹¹⁸* (*Figure 8b*, empty arrows). Actogram analysis over 30 days showed an increased number of activity peaks in the mutant groups, indicating a hyperactivity phenotype, in comparison to wild-type (*Figure 8d*). Furthermore, mean activity and sleep were also affected; the two mutant groups (*Alk^ΔRA^* and *Spar^ΔExon1^*) displayed significant variations in activity means (*Figure 8e and h–h′*; *Figure 8—figure supplement 2*). Analysis of anticipatory activity by quantifying the ratio of activity in the 3 hr period preceding light transition relative to activity in the 6 hr period preceding light transition as previously described (*Harrisingh*

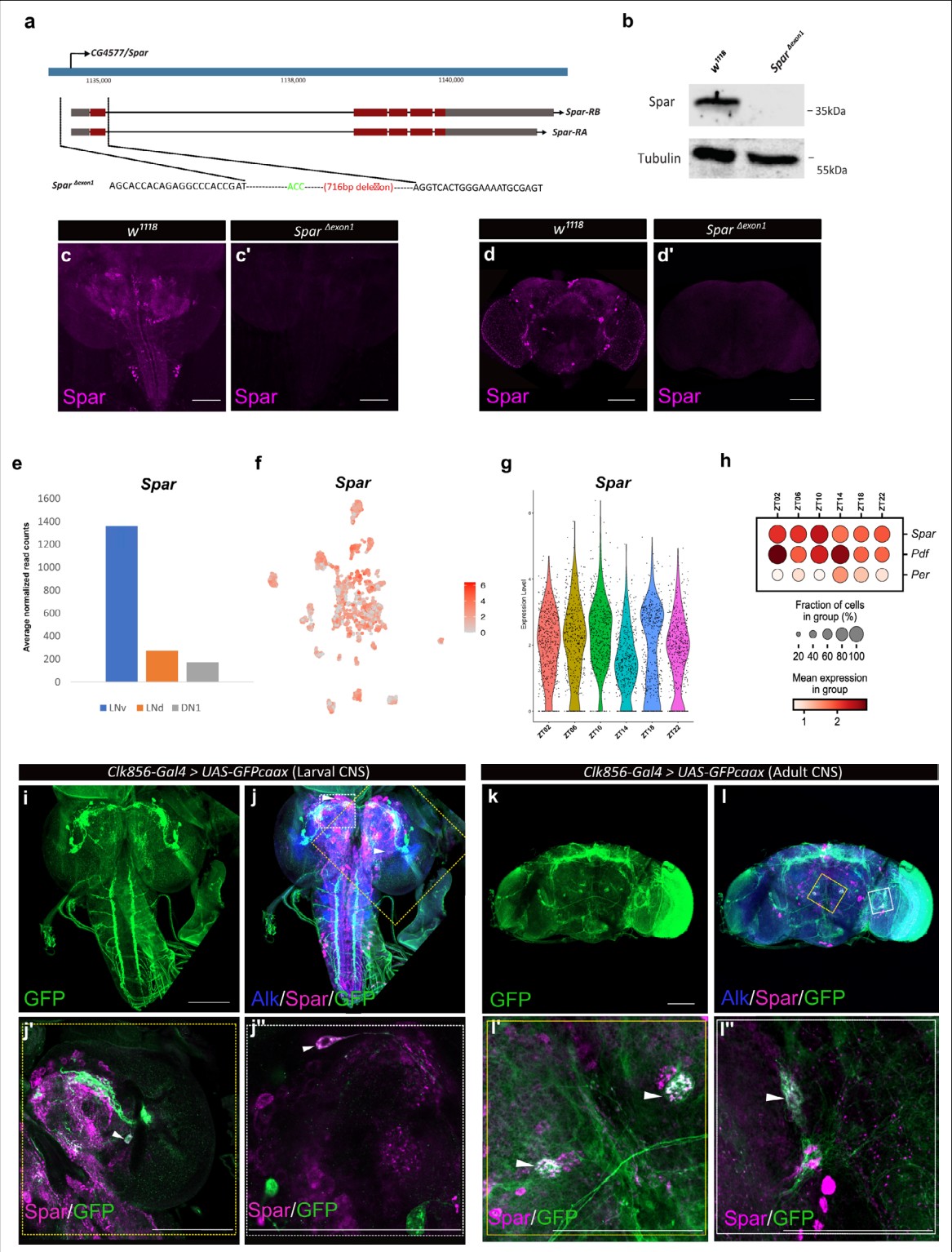

**Figure 7.** Generation of the *Spar^ΔExon1* mutant and expression of *Spar* in circadian neurons. (**a**) Schematic overview of the *Spar* gene locus and the *Spar^ΔExon1* mutant. Black dotted lines indicate the deleted region, which includes the transcriptional start and exon 1. (**b**) Immunoblotting for Spar. Spar protein (35 kDa) is present in larval central nervous system (CNS) lysates from wild-type (*w^1118^*) controls but absent in *Spar^ΔExon1* mutants. (**c–d'**) Immunostaining confirms loss of Spar protein expression in the *Spar^ΔExon1* mutant. Third instar larval (**c–c'**) and adult (**d–d"**) CNS stained for Spar (in magenta). Spar signal is undetectable in *Spar^ΔExon1*. (**e**) Expression of *Spar* in ventral lateral neuron (LNv), dorsal lateral neuron (LNd), and dorsal neuron 1 (DN1) circadian neuronal populations, employing publicly available RNA-seq data (*Abruzzi et al., 2017*). (**f**) UMAP of *Spar* expression in circadian

*Figure 7 continued on next page*

*Figure 7 continued*

neurons, employing publicly available scRNA-seq data (*Ma et al., 2021*). (**g**). Violin plot indicating *Spar* expression throughout the light-dark (LD) cycle, showing light phase (zeitgeber time 2 (ZT02), ZT06, and ZT10) and dark phase (ZT14, ZT18, and ZT22) expression. (**h**) Dot plot comparing *Spar* expression throughout the LD cycle with the previously characterized circadian-associated neuropeptide pigment dispersion factor (*Pdf*) and the core clock gene *Period* (*per*). Expression levels and percentage of expressing cells are indicated. (**i–j**) Spar expression in clock neurons (*Clk856-Gal4>UAS-GFPcaax*) of the larval CNS, visualized by immunostaining for Spar (magenta), Alk (in blue), and clock neurons (GFP, in green). (**j'–j"**) Close-up of central brain regions (yellow dashed box in **j**) indicating expression of Spar in Clk856-positive neurons (white arrowheads). (**k–l**) Immunostaining of *Clk856-Gal4>UAS-GFPcaax* in adult CNS with GFP (in green), Spar (in magenta), and Alk (in blue). (**l'–l"**) Close-ups of CNS regions (yellow dashed box in **l**) stained with GFP (in green) and Spar (in red) showing a subset of clock-positive neurons expressing Spar (white arrowheads). Scale bars: 100 μm.

The online version of this article includes the following source data and figure supplement(s) for figure 7:

**Source data 1.** Source file for immunoblotting in *Figure 7b* - raw data.

**Figure supplement 1.** Spar does not affect the Alk-regulated pupal size phenotype.

**Figure supplement 2.** *Spar* expression in circadian neuronal clusters.

*et al., 2007*) failed to identify conclusive effects on anticipatory activity in *Spar^{ΔExon1}* flies (*Figure 8f and g*). Furthermore, both *Alk^{ΔRA}* and *Spar^{ΔExon1}* exhibited significant decrease in average sleep during the day per 12 hr at young ages (days 5–7) (*Figure 8h*, *Figure 8—figure supplement 2a*). In contrast, older flies (days 20–22) did not show any significant differences in sleep patterns during the day and per 12 hr (*Figure 8h'*, *Figure 8—figure supplement 2a'*). The decrease in average sleep in both *Alk^{ΔRA}* and *Spar^{ΔExon1}* was accompanied by an increase in number of sleep bouts per 12 hr at young age (days 5–7) (*Figure 8—figure supplement 2b*) with no difference in number of sleep bouts at older age (*Figure 8—figure supplement 2b'*). Rhythmicity analysis showed that *Alk^{ΔRA}* and *w^{1118}* are more rhythmic in LD compared to *Spar^{ΔExon1}* flies (*Figure 8—figure supplement 3a*), however when comparing percentage of rhythmic flies among all groups the differences were not significant (*Figure 8—figure supplement 3a'*). Moreover, free-running period calculation by chi-square peri-odograms showed that both *w^{1118}* and *Spar^{ΔExon1}* flies exhibit a longer circadian period (higher than 1440 min), with 13% of the latter group having a shorter period (*Figure 8—figure supplement 4a–a'*). These results demonstrate that Spar is important for normal fly activity and that loss of Spar affects adult sleep/wake activity.

Since *Spar^{ΔExon1}* flies exhibited a hyperactive phenotype during both day and night hours, we sought to investigate a potential role of *Spar* in regulating the endogenous fly clock by assessing fly activity after shift to dark conditions. While control flies adapted to the LD-DD shift without any effect on mean activity and sleep, *Spar^{ΔExon1}* flies exhibited striking defects in circadian clock regulation (*Figure 9a–b'*, *Figure 8—figure supplement 1a–d'*). Comparison of average activity and sleep during 5 days of LD versus 5 days of DD (dark-dark) cycles identified a reduction in mean activity under DD conditions in *Spar^{ΔExon1}* flies (*Figure 9b–b'*). Actogram profiling showed that *Spar^{ΔExon1}* flies exhibit a hyperactive profile consistent with our previous data in LD conditions and maintain this hyperactivity when shifted into DD conditions (*Figure 9c*, *Figure 8—figure supplement 1*). Further, anticipatory peaks were largely absent on transition to DD cycle in *Spar^{ΔExon1}* mutants with no activity peaks observed at either circadian time 0 (CT0) or at CT12 (*Figure 9b*, empty arrows), consistent with a significant decrease in the a.m. and p.m. anticipatory activity (*Figure 8g*) and altered activity and sleep bouts in these mutants (*Figure 9d*, *Figure 8—figure supplement 2*). To confirm that the circadian clock activity defects observed here were specific to loss of *Spar*, we conducted a targeted knockdown of *Spar* in clock neurons, employing *Clk856-Gal4*. *Clk856-Gal4>Spar* RNAi flies exhibited a significant disruption in both activity and sleep during the DD transition period, consistent with a hyperactivity phenotype (*Figure 9e–g'*, *Figure 9—figure supplement 1*). Further comparison of *Clk856-Gal4>Spar* RNAi flies relative to controls identified a consistent increase in activity in both LD and DD conditions upon *Spar* knockdown, with a decrease in sleep observed in DD conditions (*Figure 9—figure supplement 2*). These findings agree with the expression pattern of *Spar* in clock neurons (*Figure 7*), indicating a role for Spar in circadian clock regulation. Rhythmicity analysis comparing LD and DD cycles in *Spar^{ΔExon1}* did not show a significant change indicating that *Spar^{Δexon1}* flies are mostly rhythmic in LD and DD conditions, whereas as expected, control *w^{1118}* flies were less rhythmic in DD conditions (*Figure 9—figure supplement 3a–a'*). This was also consistent when percentages of rhythmicity were deter-mined, both *w^{1118}* and *Spar^{Δexon1}* flies were rhythmic (*Figure 9—figure supplement 3b–b'*). In terms of

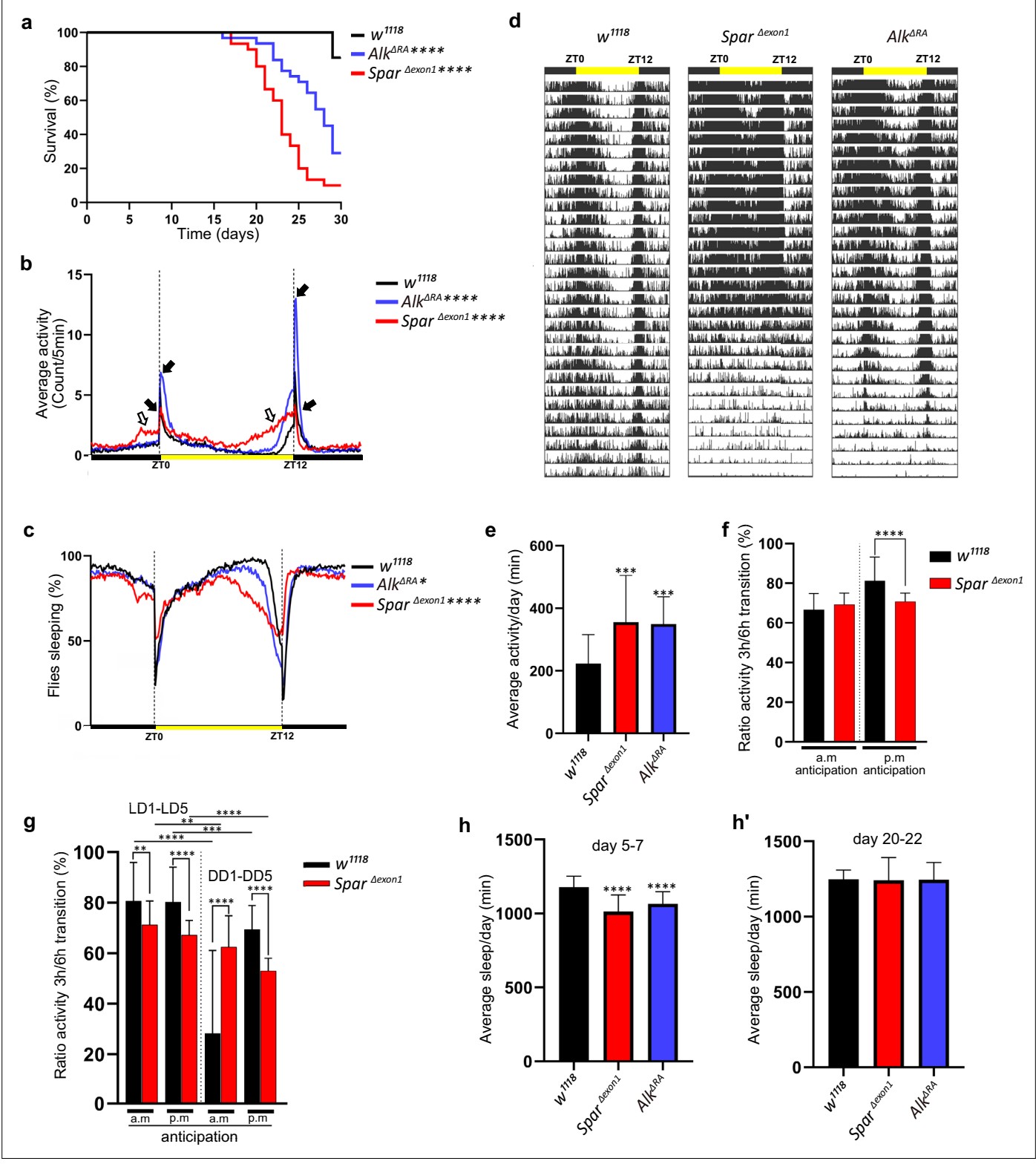

**Figure 8.** Lifespan and activity plots of *Spar^ΔExon1* mutants. (**a**) Kaplan-Meier survival curve comparing *Alk^ΔRA* (n=31) and *Spar^ΔExon1* (n=30) flies to *w^1118* controls (n=27). Outliers from each group were determined by Tukey's test, and statistical significance was analyzed by log-rank Mantel-Cox test (****p<0.0001). (**b**) Representative activity profile graph illustrating average activity count measured every 5 min across a 24 hr span. Black arrows indicate morning and evening activity peaks. Empty arrows indicate anticipatory increase in locomotor activity of *Spar^ΔExon1* mutant flies occurring before light transition. One-way ANOVA followed by Tukey's multiple comparisons post hoc test was used to determine significance between groups

*Figure 8 continued on next page*

*Figure 8 continued*

(****p<0.0001). $w^{1118}$ (n=27), $Spar^{\Delta Exon1}$ (n=30), $Alk^{\Delta RA}$ (n=31). (**c**) Representative sleep profile, demonstrating the proportion of flies engaged in sleep measured at 5 min intervals over a 24 hr period. One-way ANOVA followed by Tukey's multiple comparisons post hoc test was used to determine significance between groups (****p<0.0001; *p<0.05). $w^{1118}$ (n=27), $Spar^{\Delta Exon1}$ (n=30), $Alk^{\Delta RA}$ (n=31). (**d**) Representative average actogram of individual flies in each group. Each row corresponds to 1 day, visualized as 288 bars each representing one 5 min interval. Yellow bar represents the time of the day when the lights are turned on, with zeitgeber time 0 (ZT0) indicating the morning peak and ZT12 the evening peak. (**e**) Mean locomotor activity per day over 30 days. One-way ANOVA followed by Tukey's multiple comparisons post hoc test was used to determine significance between groups (***p<0.001). $w^{1118}$ (n=27), $Spar^{\Delta Exon1}$ (n=30), $Alk^{\Delta RA}$ (n=31). (**f**) Ratio of the mean activity in the 3 hr preceding light transition over the mean activity in the 6 hr preceding light transition. Activity data was measured over 30 days. a.m. anticipation and p.m. anticipation depict the ratio preceding lights on and lights off respectively. Unpaired Student's t-test was used to determine the significance between control and $Spar^{\Delta Exon1}$ (****p<0.0001). $w^{1118}$ (n=27), $Spar^{\Delta Exon1}$ (n=30). (**g**) Ratio of the mean activity in the 3 hr preceding light transition over the mean activity in the 6 hr preceding light transition. Activity was measured over 5 days in light-dark/dark-dark (LD/DD). a.m. anticipation and p.m. anticipation depict the ratio preceding lights on (or subjective lights on) and lights off (or subjective lights off) respectively. Unpaired Student's t-test was used to determine the significance between $Spar^{\Delta Exon1}$ and controls (****p<0.0001; **p<0.01); paired Student's t-test was used to determine significance in each group between the two experimental conditions (****p<0.0001; ***p<0.001; **p<0.01). $w^{1118}$ (n=32), $Spar^{\Delta Exon1}$ (n=31). (**h–h'**) Mean sleep per day across a 3 day average (days 5–7 (**h**), days 20–22 (**h'**)). One-way ANOVA followed by Tukey's multiple comparisons post hoc test was used to determine significance between groups (****p<0.0001). $w^{1118}$ (n=27), $Spar^{\Delta Exon1}$ (n=30), $Alk^{\Delta RA}$ (n=31). Error bars represent standard deviation.

The online version of this article includes the following figure supplement(s) for figure 8:

**Figure supplement 1.** Activity and sleep profiles of $Spar^{\Delta Exon1}$ mutants.

**Figure supplement 2.** Characterisation of sleep in $Spar^{\Delta Exon1}$ and $Alk^{\Delta RA}$ mutants.

**Figure supplement 3.** $Spar^{\Delta Exon1}$ flies retain a hyperactive profile when shifted to dark/dark conditions.

**Figure supplement 4.** Circadian period length in $Spar^{\Delta Exon1}$ mutants.

circadian period, the majority of $w^{1118}$ and $Spar^{\Delta exon1}$ flies exhibited a longer free running period in DD (*Figure 9—figure supplement 3c–d*).

## Discussion

With the advent of multiple omics approaches, data integration represents a powerful, yet challenging approach to identify novel components and targets of signaling pathways. The availability of various genetic tools for manipulating Alk signaling in *Drosophila* along with previously gathered omics dataset provides an excellent basis for Alk-centered data acquisition. We complemented this with TaDa transcriptional profiling allowing us to generate a rich dataset of Alk-responsive loci with the potential to improve our mechanistic understanding of Alk signaling in the CNS. A striking observation revealed by integrating our TaDa study with scRNA-seq data was the enrichment of Alk-responsive genes expressed in neuroendocrine cells. These results are consistent with previous studies reporting expression of Alk in the *Drosophila* larval prothoracic gland (*Pan and O'Connor, 2021*), the neuroendocrine functions of Alk in mice (*Ahmed et al., 2022*; *Reshetnyak et al., 2015*; *Witek et al., 2015*), and the role of oncogenic ALK in neuroblastoma, a childhood cancer which arises from the neuroendocrine system (*Matthay et al., 2016*; *Umapathy et al., 2019*). In this study, we focused on one target of interest downstream of Alk, however, many additional interesting candidates remain to be explored. These include *CG12594*, *complexin* (*cpx*), and the *vesicular glutamate transporter* (*VGlut*) that also exhibit a high ratio of co-expression with *Alk* in scRNA-seq data (*Figure 2—figure supplement 2*). A potential drawback of our TaDa dataset is the identification of false positives, due to non-specific methylation of GATC sites at accessible regions in the genome by Dam protein. Hence, our experimental approach likely more reliably identifies candidates which are downregulated upon Alk inhibition. In our analysis, we have limited this drawback by focusing on genes downregulated upon Alk inhibition and integrating our analysis with additional datasets, followed by experimental validation. This approach is supported by the identification of numerous previously identified Alk targets in our TaDa candidate list.

Employing a strict context-dependent filter on our integrated omics datasets identified Spar as a previously uncharacterized Alk-regulated neuropeptide precursor. Spar amino acid sequence analysis predicts an N-terminal signal peptide and multiple canonical dibasic PC cleavage sites which are hallmarks of neuropeptide precursors. These observations indicate that Spar is shuttled to the secretory pathway and is post-translationally processed within the Golgi or transport vesicles. Moreover, using

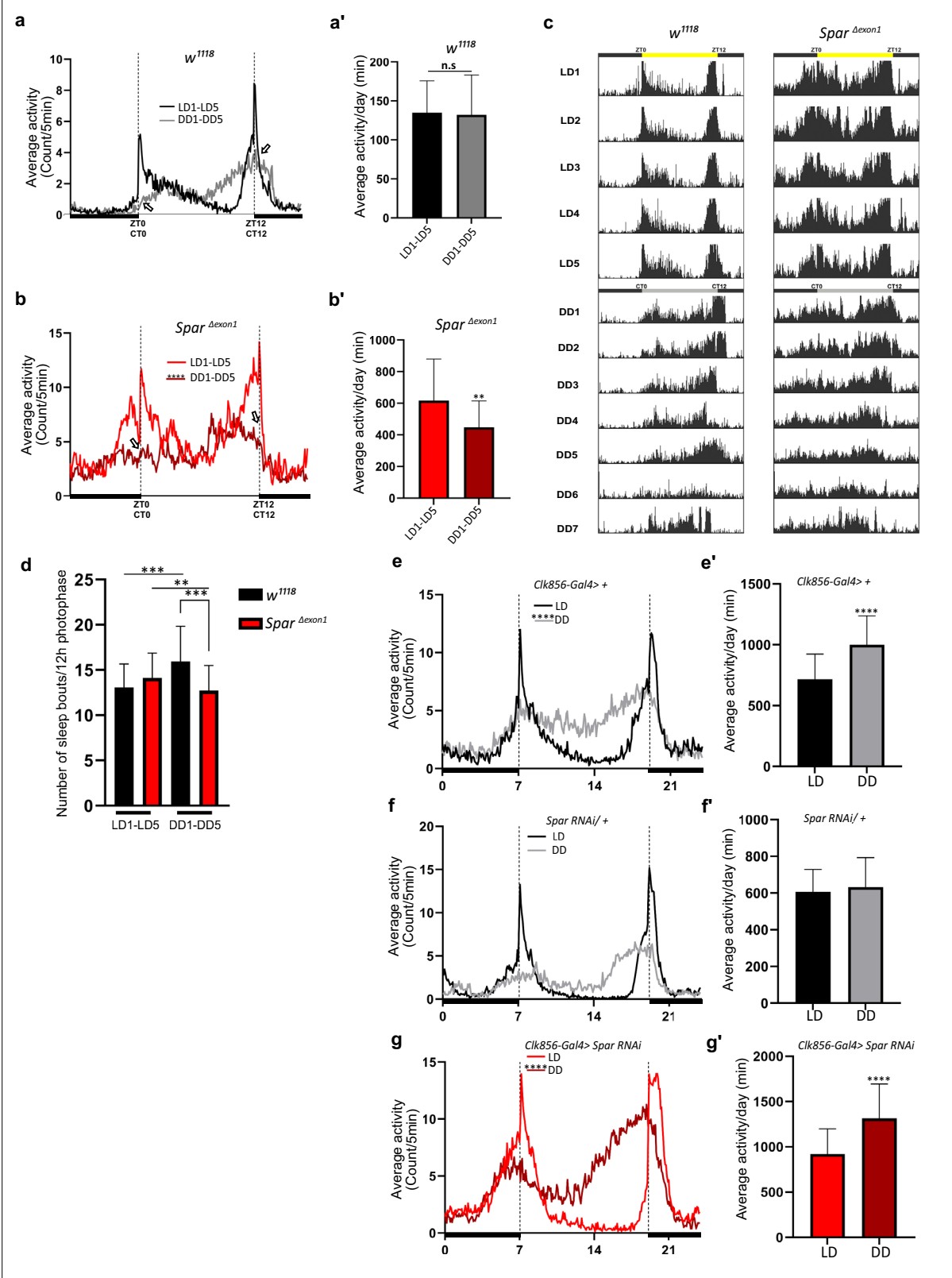

**Figure 9.** *Spar^{ΔExon1}* mutants exhibit circadian activity disturbances. (**a**) Representative activity profile for *w^{1118}* controls, illustrating the average activity count measured every 5 min across a 24 hr span for light-dark (LD) for 5 cycles (black line), subsequently switching to dark-dark (DD) for 5 cycles (gray lines). Zeitgeber time 0 (ZT0) and ZT12 represent the start and end of the photoperiod respectively. Circadian time 0 (CT0) and CT12 represent the start and end of the subjective day in constant dark conditions. Empty arrows indicate morning and evening peaks at CT0 and CT12 respectively. Paired

*Figure 9 continued on next page*

*Figure 9 continued*

Student's t-test was used to determine significance. *w^1118* (n=32). (**a'**) Mean locomotor activity per day in controls obtained by averaging 5 days in LD conditions (LD1-LD5) and 5 days in DD conditions (DD1-DD5). Paired Student's t-test was used to determine significance. *w^1118* (n=32). (**b**) Representative activity profile graph of *Spar^ΔExon1* illustrating the average activity count measured every 5 min across 24 hr obtained by averaging 5 days in LD conditions (LD1-LD5) and 5 days in DD conditions (DD1-DD5). Empty arrows indicate morning and evening peaks at CT0 and CT12 respectively. Paired Student's t-test was used to determine significance between the two experimental conditions (****p<0.0001). *Spar^ΔExon1* (n=31). (**b'**) Mean locomotor activity per day of *Spar^ΔExon1* obtained by averaging 5 days in LD conditions (LD1-LD5) and 5 days in DD conditions (DD1-DD5). Paired Student's t-test was used to determine significance (****p<0.0001). *Spar^ΔExon1* (n=31). (**c**) Representative average actograms of individual *w^1118* flies (n=32) and *Spar^ΔExon1* flies (n=31) in LD and DD conditions. Each row corresponds to 1 day, visualized in 288 bars each representing one 5 min interval. ZT0 and ZT12 represent the start and end of the photoperiod respectively. CT0 and CT12 represent the start and end of the subjective day in constant dark conditions. (**d**) Average number of sleep bouts for 12 hr photophase over 5 days in LD and the corresponding time over 5 days in DD. Unpaired Student's t-test was used to determine significance between control (*w^1118*) and *Spar^ΔExon1* (***p<0.001). Paired Student's t-test was used to determine significance between the two experimental conditions (***p<0.001; **p<0.01). *w^1118* (n=32), *Spar^ΔExon1* (n=31). (**e**) Representative activity profile graph of *Clk856-Gal4>+* illustrating the average activity count measured every 5 min across a 24 hr span for LD for 5 cycles (black line) and subsequently switching to DD for 5 cycles (gray lines). Paired Student's t-test was used to determine significance (****p<0.0001). *Clk856-Gal4>+* (n=32). (**e'**) Mean locomotor activity per day of *Clk856-Gal4>+* obtained by averaging 5 days in LD conditions (LD1-LD5) and 5 days in DD conditions (DD1- DD5). Paired Student's t-test was used to determine significance (****p<0.0001). *Clk856-Gal4>+* (n=32). *Clk856-GAL4>UAS Spar RNAi* (n=27). (**f**) Representative activity profile graph of *UAS-Spar RNAi>+* illustrating the average activity count measured every 5 min across 24 hr span obtained by averaging 5 days in LD conditions (LD1-LD5) and 5 days in DD conditions (DD1-DD5). A paired Student's t-test was used to determine the significance between the two experimental conditions. *UAS-Spar RNAi>+* (n=32). (**f'**) Graph illustrating the mean locomotor activity per day of *UAS-Spar RNAi>+* obtained by averaging 5 days in LD conditions (LD1-LD5) and 5 days in DD conditions (DD1-DD5). A paired Student's t-test was used to determine the significance between the two experimental conditions. *UAS-Spar RNAi>+* (n=32). (**g**) Representative activity profile graph of *Clk856-Gal4>UAS Spar RNAi* illustrating the average activity count measured every 5 min across 24 hr span obtained by averaging 5 days in LD conditions (LD1-LD5) and 5 days in DD conditions (DD1-DD5). Paired Student's t-test was used to determine significance (****p<0.0001). *Clk856-GAL4>UAS Spar RNAi* (n=27). (**g'**) Mean locomotor activity per day for *Clk856-Gal4>UAS Spar RNAi* obtained by averaging 5 days in LD conditions (LD1-LD5) and 5 days in DD conditions (DD1-DD5). Paired Student's t-test was used to determine the significance (****p<0.0001). Error bars represent standard deviation.

The online version of this article includes the following figure supplement(s) for figure 9:

**Figure supplement 1.** Spar^ΔExon1 mutants exhibit disturbed sleep patterns.

**Figure supplement 2.** Clock neuron specific *Spar* RNAi leads to sleep and activity disturbances.

**Figure supplement 3.** Rhythmicity and circadian period length in Spar^ΔExon1 mutants.

mass spectrometry, we were able to identify predicted canonically processed peptides from the Spar precursor in undigested fly brain extracts. While all this points toward a neuropeptide-like function of Spar, other features appear rather unusual for a typical insect neuropeptide. First, the Spar propeptide is quite large for a neuropeptide precursor, and the predicted peptides do not represent paracopies of each other and do not all carry a C-terminal amidation signal as is typical for *Drosophila* and other insect peptides (**Nässel and Zandawala, 2019**; **Wegener and Gorbashov, 2008**). Moreover, there are no obvious Spar or Spar peptide orthologs in animals outside the Diptera. We noted, however, that Spar is an acidic protein with a pI of 5.1 that lacks any cysteine residue. These features are reminiscent of vertebrate secretogranins, which are packaged and cleaved by PCs and other proteases inside dense vesicles in the regulated secretory pathway in neurosecretory cells (**Helle, 2004**). Secretogranins have so far not been identified in the *Drosophila* genome (**Hart et al., 2017**). Therefore, the identification of the neurosecretory protein Spar downstream of Alk in the *Drosophila* CNS is particularly interesting in light of previous findings, where VGF (aka secretogranin VII) has been identified as one of the strongest transcriptional targets regulated by ALK in both cell lines and mouse neuroblastoma models (**Borenäs et al., 2021**; **Cazes et al., 2014**). *VGF* encodes a precursor polypeptide, which is processed by PCs generating an array of secreted peptide products with multiple functions that are not yet fully understood at this time (**Lewis et al., 2015**; **Quinn et al., 2021**).

Using a newly generated antibody we characterized the expression of Spar in the *Drosophila* CNS, showing that its expression overlaps with the Dimm transcription factor that is expressed in the fly neuroendocrine system (**Hewes et al., 2003**), suggesting that Spar is expressed along with multiple other neuropeptides in pro-secretory cells of the CNS (**Park et al., 2008**). Spar is also expressed in well-established structures such as the mushroom bodies (**Crocker et al., 2016**), which are known to be important in learning and memory and regulate food attraction and sleep (**Joiner et al., 2006**; **Pitman et al., 2006**), and where Alk is also known to function (**Bai and Sehgal, 2015**; **Gouzi et al., 2011**; **Pfeifer et al., 2022**). Interestingly, Spar is expressed in a subset of peptidergic neurons which

emerge from the ventral nerve cord and innervate larval body wall muscle number 8. In larvae, these Lk-expressing neurons of the ventral nerve cord, known as ABLKs, are part of the circuitry that regulates locomotion and nociception, and in adults they regulate water and ion homeostasis (*Imambocus et al., 2022*; *Okusawa et al., 2014*; *Zandawala et al., 2018*). The role of Spar in this context is unknown and requires further investigation. The identity of the Spar receptor, as well as its location, both within the CNS and without, as suggested by the expression of Spar in neurons innervating the larval body wall is another interesting question for a future study. In our current study we focused on characterizing Spar in the *Drosophila* CNS. To functionally characterize Spar in this context we generated null alleles with CRISPR/Cas9 and investigated the resulting viable $Spar^{\Delta Exon1}$ mutant.

*Spar* transcript expression in *Drosophila* clock neurons has been noted in a previous study investigating neuropeptides in clock neurons, however Spar had not been functionally characterized at the time (*Abruzzi et al., 2017*; *Ma et al., 2021*). We have been able to show that Spar protein is expressed in clock neurons of the larval and adult CNS, findings that prompted us to study the effect of Spar in activity and circadian rhythms of flies. *Drosophila* activity monitoring experiments with $Spar^{\Delta Exon1}$ and Alk loss-of-function ($Alk^{\Delta RA}$) mutants revealed striking phenotypes in lifespan, activity, and sleep. In *Drosophila* a number of genes and neural circuits involved in the regulation of sleep have been identified (*Shafer and Keene, 2021*). The role of Alk in sleep has previously been described in the fly, where Alk and the Ras GTPase, Neurofibromin1 (Nf1), function together to regulate sleep (*Bai and Sehgal, 2015*). Indeed, a study in mice has reported an evolutionarily conserved role for Alk and Nf1 in circadian function (*Weiss et al., 2017*). While these studies place Alk and Nf1 together in a signaling pathway that regulates sleep and circadian rhythms, no downstream effectors transcriptionally regulated by the Alk pathway have been identified that could explain its regulation of *Drosophila* sleep/activity. Our data suggest that one way in which Alk signaling regulates sleep is through the control of Spar, as $Spar^{\Delta Exon1}$ mutants exhibit a striking activity phenotype. The role of clock neurons and the involvement of circadian input in maintenance of long-term memory (LTM) involving neuropeptides such as PDF has been previously described (*Inami et al., 2022*). Since both Alk and Nf1 are also implicated in LTM formation in mushroom body neurons (*Gouzi et al., 2018*), the potential role of Nf1 in Spar regulation and the effect of Spar loss on LTM will be interesting to test in future work. It can be noted that insulin-producing cells, DH44 cells of the pars intercerebralis, the Lk-producing LHLK neurons of the brain and certain AstA neurons in the brain are involved in regulation of aspects of metabolism and sleep (*Barber et al., 2021*; *Cavey et al., 2016*; *Chen et al., 2016*; *Cong et al., 2015*; *Donlea et al., 2018*; *Nässel and Zandawala, 2022*; *Yurgel et al., 2019*). Furthermore, the DH44 cells of the pars intercerebralis are major players in regulation of feeding and courtship in adults (*Barber et al., 2021*; *Cavanaugh et al., 2014*; *Dus et al., 2015*; *King et al., 2017*; *Oh et al., 2021*).

In conclusion, our TaDa analysis identifies a role for Alk in regulation of endocrine function in *Drosophila*. These results agree with the previously reported broad role of Alk in functions such as sleep, metabolism, and olfaction in the fly and in the hypothalamic-pituitary-gonadal axis and Alk-driven neuroblastoma responses in mice. Finally, we identify *Spar* as the first neuropeptide precursor downstream of Alk to be described that regulates activity and circadian function in the fly.

## Materials and methods

### Key resources table

| Reagent type (species) or resource | Designation | Source or reference | Identifiers | Additional information |
|---|---|---|---|---|
| Gene (*Drosophila melanogaster*) | CG4577 | FlyBase | FLYB:FBgn0031306 | Named as *Sparkly* (*Spar*) in this paper |
| Genetic reagent (*D. melanogaster*) | $w^{1118}$ | Bloomington *Drosophila* Stock Center | BDSC:3605 | |
| Genetic reagent (*D. melanogaster*) | Dimm-Gal4 | Bloomington *Drosophila* Stock Center | BDSC:25373 | Also known as *C929-Gal4* |
| Genetic reagent (*D. melanogaster*) | Clk856-Gal4 | Bloomington *Drosophila* Stock Center | BDSC:93198 | |

*Continued on next page*

*Continued*

| Reagent type (species) or resource | Designation | Source or reference | Identifiers | Additional information |
|---|---|---|---|---|
| Genetic reagent (*D. melanogaster*) | *C155-Gal4* | Bloomington *Drosophila* Stock Center | BDSC:458 | |
| Genetic reagent (*D. melanogaster*) | *UAS-Spar RNAi* | Vienna *Drosophila* Resource Center | VDRC:37830 | |
| Genetic reagent (*D. melanogaster*) | *UAS-LT3-NDam-Pol II* | *Southall et al., 2013* | | |
| Genetic reagent (*D. melanogaster*) | *UAS-Alk$^{DN}$* | *Bazigou et al., 2007* | P{UAS-Alk.EC.MYC} | |
| Genetic reagent (*D. melanogaster*) | *UAS-Jeb* | *Varshney and Palmer, 2006* | | |
| Genetic reagent (*D. melanogaster*) | *UAS-GFPcaax* | *Finley et al., 1998* | | |
| Genetic reagent (*D. melanogaster*) | *Alk$^{Y1335S}$* | *Pfeifer et al., 2022* | | |
| Genetic reagent (*D. melanogaster*) | *Alk$^{\Delta RA}$* | *Pfeifer et al., 2022* | | |
| Genetic reagent (*D. melanogaster*) | *Spar$^{\Delta Exon1}$* | This paper | | |
| Genetic reagent (*D. melanogaster*) | *UAS-Spar* | This paper | | |
| Antibody | Anti-Alk (Guinea pig polyclonal) | *Lorén et al., 2003* | | IF(1:1000) |
| Antibody | Anti-Alk (Rabbit polyclonal) | *Lorén et al., 2003* | | IF(1:1000) |
| Antibody | Anti-Alk (Guinea pig polyclonal) | *Allan et al., 2005* | | IF(1:1000) |
| Antibody | Anti-GFP (Chicken polyclonal) | Abcam | Ab13970 | IF(1:1000) |
| Antibody | Anti-PDF (Mouse polyclonal) | Developmental Studies Hybridoma Bank (DSHB) | DSHB#C7 | IF(1:1000) |
| Antibody | Anti-Ilp2 (Rabbit polyclonal) | *Veenstra et al., 2008* | | IF(1:1000) |
| Antibody | Anti-Dh44 (Rabbit polyclonal) | *Cabrero et al., 2002* | | IF(1:1000) |
| Antibody | Anti-AstA (Rabbit polyclonal) | *Stay et al., 1992*; *Vitzthum et al., 1996* Jena Bioscience GmbH | Cat#ABD-062 | IF(1:3000) |
| Antibody | Anti-Lk (Rabbit polyclonal) | *Cantera and Nässel, 1992* | | IF(1:1000) |
| Antibody | Anti-Spar (Guinea pig polyclonal) | This paper | | IF(1:2000), WB (1:1000) |
| Sequence-based reagent | sparΔExon1_F | This paper | PCR primers | caagtgaggcaattagccagaat |
| Sequence-based reagent | sparΔExon1_R | This paper | PCR primers | aacgagatgagctccgagatgg |
| Software, algorithm | Fiji | *Schindelin et al., 2012* | | |
| Software, algorithm | GraphPad Prism 8 | GraphPad Software | GraphPad Prism 8 | |

## *Drosophila* stocks and genetics

Standard *Drosophila* husbandry procedures were followed. Flies were fed on Nutri-Fly Bloomington Formulation food (Genesee Scientific, Inc) cooked according to the manufacturer's instruction. Crosses

were reared at 25°C. The following stocks were obtained from Bloomington Drosophila Stock Center (BDSC): *w^1118* (BL3605), *Dimm-Gal4* (also known as *C929-Gal4*) (BL25373), *Clk856-Gal4* (BL93198), and *C155-Gal4* (BL458). The *UAS-Spar RNAi* (v37830) line was obtained from Vienna Drosophila Resource Center. Additional stocks used in this study are the following: *UAS-LT3-NDam-Pol II* (*Southall et al., 2013*), *UAS-Alk^DN* (*P{UAS-Alk.EC.MYC}*) (*Bazigou et al., 2007*), *UAS-Jeb* (*Varshney and Palmer, 2006*), *UAS-GFPcaax* (*Finley et al., 1998*), *Alk^Y1335S* (*Pfeifer et al., 2022*), *Alk^ΔRA* (*Pfeifer et al., 2022*), *Spar^ΔExon1* (this study), *UAS-Spar* (this study).

## TaDa sample preparation

Pan neuronal *C155-Gal4* expressing animals were crossed with either *UAS-LT3-Dam::Pol II* (Control) or *UAS-LT3-Dam::Pol II; UAS-Alk^EC* (Alk dominant-negative sample) and crosses were reared at 25°C. Approximately 100–150 third instar larval brains were dissected in cold PBS for each technical replicate. Genomic DNA was extracted using a QIAGEN blood and tissue DNA extraction kit and methylated DNA was processed and amplified as previously described (*Choksi et al., 2006*; *Sun et al., 2003*) with the following modifications; after genomic DNA extraction, non-sheared gDNA was verified on 1.5% agarose gel, and an overnight DpnI digestion reaction set up in a 50 µl reaction volume. The digestion product was subsequently purified using QIAGEN MinElute PCR Purification Kit and eluted in 50 µl MQ water. 50 µl of DpnI digested and purified DNA was further used for adaptor ligation. Adaptor ligated DNA was amplified using the adaptor-specific primer to generate the TaDa-seq library. Amplified DNA from all experimental conditions was repurified (QIAGEN MinElute PCR Purification Kit) into 20 µl of MQ water and 200 ng aliquots were run on 1% agarose gel to verify amplification of TaDa library (DNA fragments ranging from 500 bp to 3 kb). The TaDa library was used for PCR-free library preparation followed by paired-end sequencing on an Illumina HiSeq 10× platform (BGI Tech Solutions, Hong Kong).

## TaDa bioinformatics data analysis

TaDa FASTQ paired-end reads of the control sample with three biological replicates and dominant-negative samples with two biological replicates (with two technical replicates for both control and dominant-negative samples) were obtained for a total of 10 samples and used for subsequent analysis. After base quality assessment, reads were mapped to the Dm6 reference genome of *D. melanogaster* using Bowtie2 (--very-sensitive-local) (*Langmead and Salzberg, 2012*) and post-alignment processes were performed with sam tools and BED tools (*Barnett et al., 2011*; *Quinlan, 2014*). The *D. melanogaster* reference sequence (FASTA) and gene annotation files were downloaded from Flybase and all GATC coordinates were extracted using fuzznuc (*Rice et al., 2000*) in BED format. Replicates were merged using Sambamba (merge) (*Tarasov et al., 2015*), and fold changes between control and dominant-negative samples, obtained by deeptools bamCompare (`--centerReads --scale-FactorsMethod readCount --effectiveGenomeSize 142573017 --smoothLength 5 -bs 1`) (*Ramírez et al., 2014*) for BIGWIG (BW) file generation. Counts of reads mapped to GATC border fragments were generated using a perl script (GATC_mapper.pl) from DamID-Seq pipeline (*Maksimov et al., 2016*). GATC level counts were converted to gene level counts using Bedtools (intersectBed) (*Quinlan, 2014*). GATC sites were merged into peaks based on a previous study (*Tosti et al., 2018*). Log2FC for individual GATC sites were generated using Limma for dominant-negative vs control (p<1e-5) and GATC sites were merged into peaks based on median GATC fragment size in the *Drosophila* genome assembly using mergeWindows (tol = 195, max.width=5000) and combineTests function from the csaw package (*Lun and Smyth, 2016*). Peaks were assigned to overlapping genes and filtered for FDR <0.05 and mean log2FC≥2. All peak calling and statistical analysis was performed using the R programming environment. TaDa data can also be visualized using a custom UCSC (University of California, Santa Cruz) Genome Browser session (https://genome-euro.ucsc.edu/s/vimalajeno/dm6). WebGestaltR (*Liao et al., 2019*) was used for GO for significantly downregulated TaDa candidates.

## Integration of TaDa data with scRNA-seq and other omics data

Previously published wild-type third instar larval brain scRNA-seq data (GSE198850) was employed (*Pfeifer et al., 2022*). Cellular heterogeneity was determined with eight different types of cells, including immature neurons, mature neurons, early neuroblast, NB-enriched cells, NB proliferating

cells, optic lobe epithelium, Repo-positive cells and Wrapper-positive cells. The mature neuron population was divided into two groups for the current study: mature neurons and neuroendocrine cells. The neuroendocrine cell cluster was determined based on canonical markers (*Guo et al., 2019*; *Hücksefeld et al., 2021*; *Nässel, 2018*; *Takeda and Suzuki, 2022*; *Torii, 2009*). Subsequent analysis, including dimensionality reduction/projection or cluster visualization, and marker identification was performed using R (Seurat) (*Stuart et al., 2019*) and Python (Scanpy) (*Wolf et al., 2018*) packages. Marker genes for each cluster were identified by FindAllMarkers function (Seurat) (*Stuart et al., 2019*). Clusters were visualized using two-dimensional Uniform Manifold Approximation and Projection (UMAP). The top 500 significantly downregulated genes from TaDa data (FDR<0.05 and mean logFC≥2) were analyzed in the third instar larval brain scRNA-seq data. These 500 candidates were used as gene signatures, and signature enrichment analysis carried out using AUCell to determine whether a subset of the input gene set was enriched for each cell (with an enrichment threshold set at >0.196), and the clusters projected in UMAP based on the signature score (AUC score) (*Aibar et al., 2017*). Violin plots, dot plots, feature plots, heatmaps, and matrix plots were used to visualize gene expression in the scRNA-seq data. Functional enrichment analysis for the common significantly downregulated genes from the TaDa analysis was compared to neuroendocrine cell markers using WebGestaltR (*Liao et al., 2019*).

## Circadian neuron scRNA-seq data analysis

Publicly available circadian neuron scRNA-seq data (10×) from the GEO database (GSE157504) was employed to investigate expression of *CG4577* in circadian neurons (*Ma et al., 2021*). The dataset includes two conditions: LD and DD, as well as six time points: 2 hr, 6 hr, 10 hr, 14 hr, and 22 hr. After preprocessing, 3172 and 4269 cells remained for the LD and DD samples respectively, with a total of 15,743 and 15,461 RNA features. Subsequent analysis, including integration, dimensionality reduction/projection, and cluster visualization, was performed using R (Seurat) (*Stuart et al., 2019*). Based on clustering, 17 clusters were defined and visualized using two-dimensional UMAP. Violin plots, dot plots, and feature plots were employed to visualize gene expression.

## Immunohistochemistry

Relevant tissue (larval CNS or body wall muscle preparation) was dissected in cold PBS and tissues fixed in 4% formaldehyde at 4°C for 1 hr. Samples were washed three times with 0.1% PBS Triton X-100, followed by overnight incubation in 4% goat serum, 0.1% PBS Triton X-100. The following primary antibodies were used: guinea pig anti-Alk (1:1000, *Lorén et al., 2003*), rabbit anti-Alk (1:1000, *Lorén et al., 2003*), and rabbit anti-Dimm (1:1000, *Allan et al., 2005*), chicken anti-GFP (1:1000, Abcam #ab13970), mouse mAb anti-PDF (1:1000, DSHB: C7), rabbit anti-Ilp2 (1:1000, *Veenstra et al., 2008*), anti-Dh44 (1:1000, *Cabrero et al., 2002*), rabbit anti-AstA (1:3000, *Stay et al., 1992*; *Vitzthum et al., 1996*, Jena Bioscience GmbH), rabbit anti-Lk (1:1000, *Cantera and Nässel, 1992*), guinea pig anti-Spar (1:2000, this study), and Alexa Fluor-conjugated secondary antibodies were from Jackson ImmunoResearch.

## Image analysis

Spar fluorescence intensity (*Figure 4i and m*) was quantified for the minimum complete confocal z-series of each third instar larval brain using Fiji (*Schindelin et al., 2012*). Confocal images from the 488 nm wavelength channel were analyzed as a Z project. Using a selection tool, Spar-positive areas were demarcated, and measurements recorded. Corrected total cell fluorescence (CTCF), in arbitrary units, was measured for each third instar brain as follows: CTCF = integrated density – (area of selected cell × mean fluorescence of background readings) (*Bora et al., 2021*; *McCloy et al., 2014*). Calculated CTCFs were represented in the form of boxplots (n=12 each for $w^{1118}$, $Alk^{Y1255S}$, $Alk^{RA}$. n=5 each for *C155-Gal4>UAS-GFPcaax* and *C155-Gal4>UAS-Alk^{DN}*, n=7 for *C155-Gal4>UAS* Jeb).

## Immunoblotting

Third instar larval brains were dissected and lysed in cell lysis buffer (50 mM Tris-Cl, pH 7.4, 250 mM NaCl, 1 mM EDTA, 1 mM EGTA, 0.5% Triton X-100, complete protease inhibitor cocktail, and PhosSTOP phosphatase inhibitor cocktail) on ice for 20 min prior to clarification by centrifugation at 14,000 rpm at 4°C for 15 min. Protein samples were then subjected to SDS-PAGE and immunoblotting

analysis. Primary antibodies used were: guinea pig anti-Spar (1:1000) (this study) and anti-tubulin (Cell Signaling #2125, 1:20,000). Secondary antibodies used were: Peroxidase Affinipure Donkey Anti-Guinea Pig IgG (Jackson ImmunoResearch #706-035-148) and goat anti-rabbit IgG (Thermo Fisher Scientific # 32260, 1:5000).

## Generation of anti-Spar antibodies

Polyclonal antibodies against Spar (CG4577) were custom generated in guinea pigs by Eurogentec. Two Spar peptides corresponding to epitopes LQEIDDYVPERRVSS (amino acids 212–226) and PVAERGSGYNGEKYF (amino acids 432–446) of Spar-PA were injected simultaneously.

## Biochemical identification of Spar peptides and phylogenetic analysis

Peptidomic data from our previous study on the role of *Drosophila* carboxypeptidase D (SILVER) in neuropeptide processing (*Pauls et al., 2019*) was re-examined for the occurrence of Spar. Peptides were extracted from brains from 5-day-old male flies and analyzed on an Orbitrap Fusion mass spectrometer (Thermo Scientific) equipped with a PicoView ion source (New Objective) and coupled to an EASY-nLC 1000 system (Thermo Scientific). Three (controls) and two (mutants) biological samples (pooled brain extracts from 30 flies) were measured in technical duplicates. The raw data is freely available at Dryad (https://doi.org/10.5061/dryad.82pr5td, for details see *Pauls et al., 2019*). Database search was performed against the UniProt *D. melanogaster* database (UP000000803; 22070 protein entries) with PEAKS XPro 10.6 software (Bioinformatics Solutions) with the following parameters: peptide mass tolerance: 8 ppm, MS/MS mass tolerance: 0.02 Da, enzyme: 'none'; variable modifications: oxidation (M), carbamidomethylation (C), pyro-glu from Q, amidation (peptide C-term). Results were filtered to 1% PSM-FDR.

To identify Spar precursor sequences in other insects and arthropods, tblastn searches with the PAM30 matrix and a low expectation threshold against the whole *Drosophila* Spar precursor or partial peptides flanked by canonical cleavage sites were performed against the NCBI databank (https://blast.ncbi.nlm.nih.gov/Blast.cgi). The obtained sequences were aligned by the MUSCLE algorithm and plotted using JalView 2 (*Waterhouse et al., 2009*).

## CRISPR/Cas9-mediated generation of the *Spar^ΔExon1^* mutant

The *Spar^ΔExon1^* mutant was generated using CRISPR/Cas9 genome editing. Design and evaluation of CRISPR target sites was performed using the flyCRISPR Optimal Target Finder tool (*Gratz et al., 2015*). Single guide RNA targeting sequences (sequences available in *Supplementary file 1*) were cloned into the pU6-BbsI-chiRNA vector (Addgene, Cat. No. 45946) and injected into *vasa-Cas9* (BDSC, #51323) embryos (BestGene Inc). Injected flies were crossed to second chromosome balancer flies (BDSC, #9120) and their progeny were PCR-screened for a deletion event. Mutant candidates were confirmed by Sanger sequencing (Eurofins Genomics).

## Generation of *UAS-Spar* fly lines

*UAS-Spar* was generated by cloning (GeneScript) the coding sequence of *CG4577-RA* into EcoRI/XbaI-cut *pUASTattB* vector followed by injection into fly embryos (BestGene Inc) using attP1 (second chromosome, BDSC#8621) and attP2 (third chromosome, BDSC#8622) docking sites for phiC31 integrase-mediated transformation. Injected flies were crossed to second or third chromosome balancer flies, and transgenic progeny identified based on the presence of mini-white marker.

## Measurement of pupal size

Late pupae of the indicated genotype were collected and placed on glass slides with double-sided tape. Puparium were imaged with a Zeiss Axio Zoom.V16 stereo zoom microscope with a light-emitting diode ring light and measured using Zen Blue edition software. Both female and male pupae, picked randomly, were used for measurements.

## *Drosophila* activity monitor assay

Up to 32 newly eclosed male flies were transferred into individual glass tubes containing food media (1% agar and 5% sucrose), which were each placed into a DAM2 *Drosophila* activity monitor (Trikinetics Inc). Monitors were then placed in a 25°C incubator running a 12:12 hr LD cycle, at a constant

60% humidity. Activity was detected by an infrared light beam emitted by the monitor across the center of each glass tube. The experiment was carried out for 1 month, and the raw binary data was acquired by the DAMSystem310 software (Trikinetics Inc). The LD/DD experiment was performed according to previously published work (*Chiu et al., 2010*); adult flies were first entrained for 5 days in normal LD cycle and on the last day (LD5), the light parameters were switched off and flies were then conditioned in complete DD settings for 7 days. Raw data analysis was carried out using a Microsoft Excel macro (*Berlandi et al., 2017*) taking into consideration 5 min of inactivity as sleep and more than 24 hr of immobility as a death event. The activity and sleep parameters are calculated for each day of the experiment as average from data of all living animals at this time and are displayed over the duration of the experiment. Calculation of anticipatory activity was performed accordingly to previously published work (*Harrisingh et al., 2007*) by quantifying the ratio of activity in the 3 hr preceding light transition to activity in the 6 hr preceding light transition, defined as a.m and p.m anticipation for the 6 hr period before lights on and 6 hr period before lights off, respectively. Actogram activity profile charts were generated using ActogramJ 1.0 (https://bene51.github.io/ActogramJ/index.html) and ImageJ software (https://imagej.nih.gov/ij/). ActogramJ was further used to generate the chi-square periodogram for each single fly in order to calculate the power value of rhythmicity and the percentage of rhythmic flies. All statistical analysis were performed using GraphPad Prism 8.4.2.

## Data visualization and schematics

Schematics were generated at Biorender.com and Bioicons.com. The pipeline icon by Simon Dürr (https://twitter.com/simonduerr) is licensed under CC0 (https://creativecommons.org/publicdomain/zero/1.0/). Boxplots in *Figure 3d* and *Figure 7—figure supplement 1* were generated using BoxplotR (http://shiny.chemgrid.org/boxplotr/). Boxplots in *Figure 4* were generated using GraphPad Prism 9.

## Acknowledgements

The authors thank Jonathan Benito Sipos and Stefan Thor for the kind gift of anti-Dimmed antibodies, as well as Jan Veenstra for kindly gifting anti-Dh44 and anti-Ilp2. C7 anti-PDF (developed by J Blau) was obtained from the Developmental Studies Hybridoma Bank, created by the NICHD of the NIH and maintained at The University of Iowa, Department of Biology, Iowa City, IA 52242. We acknowledge Bloomington Drosophila Stock Center (NIH P40OD018537) for fly stocks used in this study. We thank Hisae Mori for providing support for fly lab maintenance. We thank members of the Palmer, Hallberg lab, and Anne Uv for critical feedback on the manuscript. We thank Bengt Hallberg for access to the premium version of Biorender.com. This work has been supported by grants from the Swedish Cancer Society (RHP CAN21/01549), the Children's Cancer Foundation (RHP 2019-0078), the Swedish Research Council (RHP 2019-03914), the Swedish Foundation for Strategic Research (RB13-0204), the Göran Gustafsson Foundation (RHP2016), the Knut and Alice Wallenberg Foundation (KAW 2015.0144), and Assar Gabrielsson's foundation (SKS FB23-104). MS and JS are supported by the Medical Practice Plan (MPP) at the American University of Beirut.

## Additional information

### Funding

| Funder | Grant reference number | Author |
| --- | --- | --- |
| Swedish Cancer Foundation | CAN21/01549 | Ruth H Palmer |
| Barncancerfonden | RHP 2019-0078 | Ruth H Palmer |
| Vetenskapsrådet | RHP 2019-03914 | Ruth H Palmer |
| Stiftelsen för Strategisk Forskning | RB13-0204 | Ruth H Palmer |
| Göran Gustafsson Foundation | RHP2016 | Ruth H Palmer |

| Funder | Grant reference number | Author |
|--------|------------------------|--------|
| Knut och Alice Wallenbergs Stiftelse | KAW 2015.0144 | Ruth H Palmer |
| Stiftelsen Assar Gabrielssons Fond | FB23-104 | Sanjay Kumar Sukumar |
| American University of Beirut | MPP | Margret Shirinian Jawdat Sandakly |

The funders had no role in study design, data collection and interpretation, or the decision to submit the work for publication.

## Author contributions

Sanjay Kumar Sukumar, Conceptualization, Formal analysis, Supervision, Validation, Investigation, Visualization, Methodology, Writing - original draft, Writing – review and editing; Vimala Antony-dhason, Resources, Data curation, Software, Formal analysis, Investigation, Writing – review and editing; Linnea Molander, Validation, Investigation, Writing – review and editing; Jawdat Sandakly, Formal analysis, Validation, Investigation, Visualization, Methodology, Writing – review and editing; Malak Kleit, Ganesh Umapathy, Formal analysis, Investigation, Visualization, Writing – review and editing; Patricia Mendoza-Garcia, Conceptualization, Investigation, Methodology, Writing – review and editing; Tafheem Masudi, Formal analysis, Investigation, Visualization, Methodology, Writing – review and editing; Andreas Schlosser, Formal analysis, Investigation, Methodology, Writing – review and editing; Dick R Nässel, Conceptualization, Writing – review and editing; Christian Wegener, Margret Shirinian, Data curation, Formal analysis, Supervision, Funding acquisition, Investigation, Visualization, Methodology, Writing – review and editing; Ruth H Palmer, Conceptualization, Formal analysis, Supervision, Funding acquisition, Investigation, Writing - original draft, Project administration, Writing – review and editing

## Author ORCIDs

Sanjay Kumar Sukumar (iD) http://orcid.org/0000-0001-9543-3113
Vimala Antonydhason (iD) http://orcid.org/0000-0001-6982-4030
Jawdat Sandakly (iD) http://orcid.org/0000-0002-5739-793X
Malak Kleit (iD) http://orcid.org/0009-0004-2039-3507
Ganesh Umapathy (iD) http://orcid.org/0000-0003-2324-8300
Patricia Mendoza-Garcia (iD) http://orcid.org/0000-0002-6084-7962
Tafheem Masudi (iD) http://orcid.org/0000-0001-7406-9084
Andreas Schlosser (iD) http://orcid.org/0000-0003-0612-9932
Dick R Nässel (iD) http://orcid.org/0000-0002-1147-7766
Christian Wegener (iD) http://orcid.org/0000-0003-4481-3567
Margret Shirinian (iD) http://orcid.org/0000-0003-4666-2758
Ruth H Palmer (iD) http://orcid.org/0000-0002-2735-8470

Reviewer #2 (Public Review): https://doi.org/10.7554/eLife.88985.5.sa1
Reviewer #3 (Public Review): https://doi.org/10.7554/eLife.88985.5.sa2
Author response https://doi.org/10.7554/eLife.88985.5.sa3

## Additional files

### Supplementary files

• Supplementary file 1. Targeted DamID (TaDa) data Alk$^{DN}$ downregulated genes (Sheet 1). RNA-seq normalized read count data of *CG4577* in control (*w$^{1118}$*), *Alk$^{RA}$*, and *Alk$^{Y1355S}$* conditions (Sheet 2). RNA-seq average normalized read count data of *Spar* in ventral lateral neuron (LNv), dorsal lateral neuron (LNd), and dorsal neuron 1 (DN1) clock neuronal cells (Sheet 3). *Spar$^{\Delta Exon1}$* mutant CRISPR single guide RNA target and screening primer information (Sheet 4).

• MDAR checklist

## Data availability

The original contributions presented in the study are included in the article/Supplementary Material. The TaDa dataset has been deposited in Gene Expression Omnibus (GEO) under the accession number GSE229518. The genome browser tracks for the TaDa peak analysis can be found at: https://genome-euro.ucsc.edu/s/vimalajeno/dm6.

The following dataset was generated:

| Author(s) | Year | Dataset title | Dataset URL | Database and Identifier |
|---|---|---|---|---|
| Palmer R, Anthonydhason V, Sukumar SK | 2024 | The Alk receptor tyrosine kinase regulates Sparkly, a novel activity regulating neuropeptide precursor in the *Drosophila* CNS | https://www.ncbi.nlm.nih.gov/geo/query/acc.cgi?acc=GSE229518 | NCBI Gene Expression Omnibus, GSE229518 |

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
