## [Editor Report · eLife assessment]

This paper characterises a novel gene (*Spar*), and presenting **valuable** findings in the field of insect biology and behaviour. The experiments are well designed, with attention to detail, showcasing the potential of the *Drosophila melanogaster* model and the use of online resources. The mixed approach presents a **convincing** argument for a genetic interaction between Alk and *Spar*.

---

## [Referee Report · Reviewer #2 (Public Review)]

This manuscript illustrates the power of "combined" research, incorporating a range of tools, both old and new to answer a question. This thorough approach identifies a novel target in a well-established signalling pathway and characterises a new player in *Drosophila* CNS development.

Largely, the experiments are carried out with precision, meeting the aims of the project, and setting new targets for future research in the field. It was particularly refreshing to see the use of multi-omics data integration and Targeted DamID (TaDa) findings to triage scRNA-seq data. Some of the TaDa methodology was unorthodox, however, this does not affect the main finding of the study. The authors (in the revised manuscript) have appropriately justified their TaDa approaches and mentioned the caveats in the main text.

Their discovery of Spar as a neuropeptide precursor downstream of Alk is novel, as well as its ability to regulate activity and circadian clock function in the fly. Spar was just one of the downstream factors identified from this study, therefore, the potential impact goes beyond this one Alk downstream effector.

---

## [Referee Report · Reviewer #3 (Public Review)]

Summary:

The receptor tyrosine kinase Anaplastic Lymphoma Kinase (ALK) in humans is nervous system expressed and plays an important role as an oncogene. A number of groups have been studying ALK signalling in flies to gain mechanistic insight into its various roles. In flies, ALK plays a critical role in development, particularly embryonic development and axon targeting. In addition, ALK was also shown to regulate adult functions including sleep and memory. In this manuscript, Sukumar et al., used a suite of molecular techniques to identify downstream targets of ALK signalling. They first used targeted DamID, a technique that involves a DNA methylase to RNA polymerase II, so that GATC sites in close proximity to PolII binding sites are marked. They performed these experiments in wild type and ALK loss of function mutants (using an Alk dominant negative ALkDN), to identify Alk responsive loci. Comparing these loci with a larval single cell RNAseq dataset identified neuroendocrine cells as an important site of Alk action. They further combined these TaDa hits with data from RNA seq in Alk Loss and Gain of Function manipulations to identify a single novel target of Alk signalling - a neuropeptide precursor they named Sparkly (Spar) for its expression pattern. They generated a mutant allele of Spar, raised an antibody against Spar, and characterised its expression pattern and mutant behavioural phenotypes including defects in sleep and circadian function.

Strengths:

The molecular biology experiments using TaDa and RNAseq were elegant and very convincing. The authors identified a novel gene they named Spar. They also generated a mutant allele of Spar (using CrisprCas technology) and raised an antibody against Spar. These experiments are lovely, and the reagents will be useful to the community. The paper is also well written, and the figures are very nicely laid out making the manuscript a pleasure to read.

Weaknesses:

The manuscript has improved very substantially in revision. The authors have clearly taken the comments on board in good faith.

Editors' note: The authors have satisfactorily addressed the concerns raised in the previous rounds of review. These were related to the unconventional analysis of the TaDa data, the addition of other means of down regulated gene function, and the nature of analyses of behavioural data.

---

## [Author Response]

The following is the authors’ response to the previous reviews.

Point-by-point response to concerns raised by reviewer #3:

The manuscript has improved very substantially in revision. The authors have clearly taken the comments on board in good faith. Yet, some small concerns remain around the behavioural analysis.In Fig. 8H and H' average sleep/day is ~100. Is this minutes of sleep? 100 min/day is far too low, is it a typo?The numbers for sleep bouts are also too low to me e.g. in Fig 9 number of sleep bouts avg around 4.In their response to reviewers the authors say these errors were fixed, yet the figures appear not to have been changed. Perhaps the old figures were left in inadvertently?

Indeed this correction was somehow missed and we thank the reviewer for noticing this. We have now corrected Fig 8H-H’ and Fig 9D.

The circadian anticipatory activity analyses could also be improved. The standard in the field is to perform eduction analyses and quantify anticipatory activity e.g. using the method of Harrisingh et al. (PMID: 18003827). This typically computed as the ratio of activity in the 3hrs preceding light transition to activity in the 6hrs preceding light transition.In their response to reviewers, the authors have revised their anticipation analyses by quantifying the mean activity in the 6 hrs preceding light transition. However, in the method of Harrisingh et al., anticipation is the ratio of activity in the 3hrs preceding light transition to activity in the 6hrs preceding light transition. Simply computing the activity in the 6hrs preceding light transition does not give a measure of anticipation, determining the ratio is key.

We acknowledge the importance of obtaining accurate results in our analysis, therefore we have re-evaluated the anticipation activity by measuring the ratio of the mean activity in the 3h preceding light transition over the activity in the 6h preceding light transition. We have reported the data as percentages in Fig 8F-G and modified the figure legends accordingly.